

# Quantification of the radiative impact of light-absorbing particles during two contrasted snow seasons at Col du Lautaret (2058 m a.s.l., French Alps)

Francois Tuzet[1,2], Marie Dumont[1], Ghislain Picard[2], Maxim Lamare[1], Didier Voisin[2], Pierre Nabat[1], Mathieu Lafaysse[1], Fanny Larue[1], Jesus Revuelto[1,3], and Laurent Arnaud[2]

[1]Univ. Grenoble Alpes, Université de Toulouse, Météo-France, CNRS, CNRM, Centre d'Etudes de la Neige, 38000 Grenoble, France
[2]UGA,CNRS, Institut des Geosciences de l'Environnement (IGE) UMR 5001, Grenoble, France
[3]Instituto Pirenaico de Ecología, Consejo Superior de Investigaciones Científicas (IPE-CSIC), Zaragoza, Spain

**Correspondence:** Marie Dumont (marie.dumont@meteo.fr)

**Abstract.** The presence of light-absorbing particles (LAPs) in snow leads to a decrease in shortwave albedo, affecting the surface energy budget. Precisely quantifying the impacts of LAPs on snowpack evolution is crucial to characterise the spatio-temporal variability of snowmelt and assess snow albedo feedbacks in detail. However, the understanding of the impacts of LAPs is hampered by the lack of dedicated datasets, as well as the scarcity of models able to represent the interactions between LAPs and

snow metamorphism. The present study aims to address both these limitations by introducing a survey of LAP concentrations over two snow seasons in the French Alps, as well as an estimation of their impacts based on the Crocus snowpack model that represents the complex interplays between LAP dynamics and snow metamorphism.

First, we present a unique dataset collected at the Col du Lautaret (2058 m a.s.l; French Alps) for the two snow seasons 2016–2017 and 2017–2018. This dataset consists of spectral albedo measurements (manual and automated), vertical profiles of

snow specific surface area (SSA), density, and concentrations of refractive Black Carbon (rBC), Elemental Carbon (EC) and mineral dust. Spectral albedo data are processed to estimate near-surface SSA and LAP absorption-equivalent concentrations near the surface of the snowpack. These estimates are then compared to chemical measurements of dust and BC concentrations, as well as to SSA measurements acquired by near-infrared reflectometry. Our dataset highlights large discrepancies between the two measurement techniques of BC concentrations, with EC concentrations being one order of magnitude higher than rBC

measurements. In view of LAP absorption inferred from albedo measurements, the mass absorption efficiency (MAE) of BC used in our study (11.25 g m$^{-2}$ at 550 nm) is more appropriate for EC measurements than for rBC ones.

Second, we present ensemble snowpack simulations of ESCROC – the multi-physics version of the detailed snowpack model Crocus – forced with in-situ meteorological data as well as dust and BC deposition fluxes from the ALADIN-Climate atmospheric model. The results of these simulations are compared to the near-surface properties estimated from automatic albedo

measurements, showing that the temporal variations of near-surface LAP concentration and SSA are correctly reproduced. The





impact of dust and BC on our simulations is estimated by comparing this ensemble to a similar ensemble that does not account for LAPs. The seasonal radiative forcing of LAPs is 1.33 times higher for the 2017–2018 snow season than for the 2016–2017 one, highlighting a strong variability between these two seasons. However, the shortening of the snow season caused by LAPs are similar with $10\pm5$ and $11\pm1$ days for the first and the second snow seasons respectively. This counter-intuitive result is attributed

5  to two small snowfalls at the end of the first season and highlights the importance to account for meteorological conditions to correctly predict the impact of LAPs. The strong variability of season shortening caused by LAPs in the multi-physics ensemble for the first season also points out the sensitivity of model-based estimations of LAP impact to modelling uncertainties of other processes. Finally, the indirect impact of LAPs (i.e. the enhancement of energy absorption due to acceleration of the metamorphism by LAPs) is negligible for the two years considered here, contrary to what was found in previous studies for

10  other sites. This finding is mainly attributed to the meteorological conditions of the two studied snow seasons.

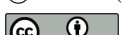



## 1 Introduction

Light-absorbing particles (LAPs) such as black carbon (BC) or mineral dust (hereinafter referred to as dust) are important drivers of snow albedo (Warren and Wiscombe, 1980). Indeed, LAPs enhance solar energy absorption in the visible (direct impact), triggering changes in snow properties that further decrease albedo (indirect impacts; e.g. Hansen and Nazarenko, 2004; Painter

et al., 2007; Skiles and Painter, 2019). As a consequence, LAPs have a strong effect on snowpack evolution and melt at global and local scales. Since LAPs are present in snow-covered regions worldwide, they have been reported as a powerful climate forcing parameter (Flanner et al., 2007). In some regions, intensive field campaigns have already been conducted, providing a thorough knowledge of local or regional impacts of LAPs (e.g. Doherty et al. (2010) in the Arctic or Skiles et al. (2012) and Painter et al. (2012) in the Rocky Mountains; USA). For instance, dust deposition in the Rocky Mountains was shown to shorten

the presence of snow cover by up to 51 days (Skiles et al., 2012). However, the impact of LAPs varies widely at the regional scale, and the recent review from Skiles et al. (2018) recommends to expand local-scale LAP observations.

In the European Alps, the only multi-year records of LAP concentrations are provided by ice cores for high altitude glaciers located mostly above 4000 m. In the 1980s, the analysis of ice cores from several alpine glaciers showed the importance of Saharan dust and BC in this region (De Angelis and Gaudichet, 1991; Thevenon et al., 2009). Gabbi et al. (2015) recently

estimated that the mean LAP radiative forcing (RF; i.e the enhancement of shortwave radiation absorption caused by LAPs) at Claridenfirn (Swiss Alps) increased by 3.2 W m$^{-2}$ due to BC and 0.6 W m$^{-2}$ due to dust between 1914 and 2014. De Angelis and Gaudichet (1991) have shown an increasing trend in Saharan dust deposition on glaciers in the French Alps between 1955 and 1985. The increase of extreme Saharan dust deposition events has also recently been confirmed and partly attributed to the Arctic amplification of global warming (Varga et al., 2019). Other ice core analyses in the Alps point out the increase of BC

concentration from 1850-1870 to the middle of the 20th century owing to industrialisation (Thevenon et al., 2009; Jenk et al., 2006; Painter et al., 2013).

For seasonal snow, measurements of LAP concentrations in the European Alps are scarcer than long term measurements from ice cores. Such measurements are however essential to gather information about the seasonal evolution of LAPs at lower altitudes. Di Mauro et al. (2015) report detailed measurements of dust concentrations using both a particle counter and gravimetry

techniques, but only for samples taken on a single day. A similar dataset is presented in Di Mauro et al. (2019) with one day of dust measurements after a strong deposition event. The longest monitoring program of LAP concentrations in the European Alps is presented by Dumont et al. (2017), covering two months at the end of a snow season at a mid-altitude site (1300 m). To our knowledge, no other intensive LAP monitoring survey has been conducted so far on seasonal snow in the Alps, which strongly limits the understandings of their impacts.

There are mainly two experimental approaches to determine the radiative impact of LAPs in seasonal snow. First, several types of chemical measurement techniques were developed to estimate the concentration of different LAP species in snow. Once the concentration determined, it can be related to LAP absorption under an assumption of the LAP mass absorption efficiency (MAE), i.e. the absorption efficiency of the LAP by unit of mass (g m$^{-1}$). Nevertheless, the MAE values for LAPs in snow are poorly constrained due to several problems such as the coating of LAP particles (e.g. Dong et al., 2018) or their mixing-state in



relation to the ice-matrix (e.g. Flanner et al., 2012). Second, an alternative approach consists of using spectral measurements of snow reflectance to infer the radiative impact of LAPs in snow. The method was developed to circumvent the difficulties associated with chemical measurements as well as the aforementioned MAE uncertainties.

Major efforts have been devoted to chemically determining BC concentrations in the atmosphere over the past decades (e.g.

Bond and Bergstrom, 2006), all leading to problematic results. Indeed, different experimental methods have been developed, taking advantage of different physical properties of BC aerosols, yet no consensus has ever been reached on a preferential technique to measure BC (e.g. Petzold et al., 2013). For this reason, Petzold et al. (2013) defined a specific terminology for reporting BC measurements, where refractory Black Carbon (rBC) refers to BC measured by laser-induced incandescence (e.g.; Schwarz et al., 2008), Elemental Carbon (EC) to methods based on evolved carbon (e.g. Zanatta et al., 2016) and equivalent

Black Carbon (eBC) to methods based on light absorption (e.g. Bond et al., 1999). Strong discrepancies between these different methods are observed. Watson (2005) presents a review of numerous inter-comparisons of BC measurements techniques, highlighting up to 7-fold differences, which lead to a 1 order of magnitude uncertainty in experimental MAE estimations. Moreover, measurement techniques of BC concentrations used for the atmosphere may not be directly transferable to snow. For instance, the size distribution of BC is suspected to be shifted towards bigger particles in snow compared to that in the

atmosphere (e.g. Lim et al., 2014; Schwarz et al., 2013), affecting amongst others the influence of the size detection range on the measurement. Additional measurement uncertainties due to the liquid state of the samples are also expected (e.g. nebulisation biases, Schwarz et al., 2012). Strong uncertainties are therefore associated with BC measurements in snow. The concentration of dust in snow can also be measured with various chemical techniques, for instance using particle counters (e.g. Coulter counter in Delmonte et al., 2004), by gravimetry (e.g. Di Mauro et al., 2015), or based on dust mineralogical properties (e.g.

De Angelis and Gaudichet, 1991). As the attention paid to dust has been hitherto lower than BC, knowledge of uncertainties of dust concentration measurements in snow is incomplete.

The second approach consists in analysing measurements of snow optical properties to infer the absorption caused by LAPs, exploiting the differences between LAP and ice absorption spectra. To isolate the absorption of LAPs, these methods often rely on spectral band ratios (e.g. Kokhanovsky et al., 2018) and/or on a radiative transfer model (e.g. Dumont et al., 2017; Lamare

et al., 2016; Tuzet et al., 2019). The inferred absorption can then be used in two ways. First, assumptions on the incoming radiation make it possible to deduce the RF of LAPs from spectral reflectance measurements (e.g. Painter et al., 2007; Skiles et al., 2012). Second, an assumption on the MAE value of LAPs allows to estimate an Absorption equivalent concentration (AEC) from spectral albedo measurements, i.e. the LAP concentration that would cause the same absorption as observed in albedo measurements. For instance, Dumont et al. (2017) derive temporal series of near-surface AEC from automated spectral

albedo measurements. Similar methods have also been applied to data acquired by unmanned aerial vehicles (e.g. Di Mauro et al., 2015) or satellites (e.g. Kokhanovsky et al., 2019). Recently, a method was proposed to estimate AECs of homogeneous snow layers in depth from vertical profiles of spectral irradiance (Tuzet et al., 2019).

A variety of snow radiative transfer models accounting for the impact of LAPs were developed in the last decades (e.g. Warren and Wiscombe, 1980; Stamnes et al., 1988; Flanner and Zender, 2005; Flanner et al., 2007; Aoki et al., 2011; Libois

et al., 2013). The direct RF of LAPs can be computed by simulating snow albedo with and without LAPs, using measured or





simulated LAP concentrations (e.g. Hadley and Kirchstetter, 2012), and thus only requires snow radiative transfer modelling. In contrast, estimating the indirect RF of LAPs – which accounts for the albedo feedbacks, i.e the interaction between LAP impacts and snow metamorphism – necessitates a coupling between a radiative transfer model and a snowpack model simulating snow metamorphism. Tuzet et al. (2017) introduced an explicit representation of LAP deposition and evolution in the Crocus detailed

snowpack model (Brun et al., 1989; Vionnet et al., 2012), in which the recent implementation of the Two-stream Analytical Radiative TransfEr in Snow model (TARTES, Libois et al., 2013) allows to simulate spectral albedo. These developments make it possible to simulate the indirect impact of LAPs. Skiles and Painter (2019) similarly coupled the snowpack evolution model SNOWPACK (Lehning et al., 2002) and the radiative transfer model SNICAR (SNow, Ice and Aerosol Radiation model; (Flanner and Zender, 2005)). Tuzet et al. (2017) and Skiles and Painter (2019) estimated that the indirect RF is an efficient

mechanism of the RF of LAPs, accounting for 15 to 20 % of the total RF of LAPs. However, detailed snowpack models are affected by many uncertainties coming either from the uncertainties in the atmospheric forcing or from intrinsic uncertainties in the representation of snow physics (modelling uncertainties; e.g. Krinner et al., 2018, Raleigh et al., 2015, Essery et al., 2013). Accounting for these uncertainties is of particular interest since the errors accumulate over time, leading to strong uncertainties at the end of the snow season. Modelling uncertainties have recently been shown to affect the conclusions drawn on the impacts

of LAPs using a model-based approach. Indeed, Skiles and Painter (2019) demonstrated that the estimated shortening of the snow season caused by LAPs varies from 30 to 49 days depending on the complexity of the snowpack model used. In this light, a multi-physics ensemble modelling framework called ESCROC (Ensemble System CROCus; Lafaysse et al., 2017) has been developed for Crocus to represent its own modelling uncertainties. The combined use of this ensemble modelling framework and developments of Tuzet et al. (2017) make it possible to represent all the impacts (direct and indirect) of LAPs as well as the

modelling uncertainties associated with the other process in the snowpack model.

The present study aims to answer two scientific questions:

1) What are the concentrations of BC and dust near the surface of an Alpine snowpack and how do they evolve over two snow seasons?

2) What is the impact of these LAPs on the snowpack evolution and more specifically on snow cover duration?

To answer the first question, two years of near-weekly measurements were performed at the Col du Lautaret study site during which the impact of LAPs was determined by both aforementioned approaches. This unique dataset, presented in Section 2, consists of 30 measurement days during the 2016–2017 and 2017–2018 seasons. The spectral albedo measurements are first processed to estimate snowpack near-surface specific surface area (SSA) and AEC as described in Section 3. These data are then compared to physico-chemical measurements performed in snowpits.

To address the second question, the impact of LAPs on snowpack evolution is then calculated using ensemble simulations with the multiphysics version of Crocus model (ESCROC; Lafaysse et al., 2017) including TARTES. Two ensemble simulations accounting or not for the impact of LAP are run and compared to each other, providing an estimate of the impact of LAPs on snowpack evolution and the associated uncertainty. The results of our analysis are presented (Section 4) and discussed (Section 5) before conclusions are drawn (Section 6).





**Summary of the acronyms used in the present study**

| Acronym | Full name |
| --- | --- |
| AEC | Absorption equivalent concentration |
| BC | Black carbon |
| eqBC | Equivalent black carbon |
| rBC | Refractory black carbon |
| eqEC | Equivalent elemental carbon |
| eqrBC | Equivalent refractory black carbon |
| DOY | Day of year |
| EC | Elemental carbon |
| ESCROC | Ensemble system Crocus |
| LAP | Light-absorbing particle |
| MAE | Mass absorption efficiency |
| RF | Radiative forcing |
| ROS | Rain on snow |
| RMSE | Root mean squared error |
| SSA | Specific surface area |
| TARTES | Two-stream Analytical Radiative TransfEr in Snow |

## 2 Materials

Measurements were collected during the two snow seasons 2016–2017 and 2017–2018 at the Col du Lautaret site (45°02'28.7"N,
6°24'38.0"E) around 2100 m a.s.l. in the French Alps (Figure 1). This site was chosen due to its high elevation and easy
accessibility, even during the snow season. The dataset includes automated measurements and manual measurements, the latter
acquired over 30 days during the two seasons. The following variables were collected: 1) Automated and manual spectral albedo
measurements, 2) Snow physical properties from a snowpit including: snow depth, snow water equivalent as well as vertical
profiles of SSA, density, temperature and snow type, 3) Vertical profiles of EC, rBC and dust chemical concentrations and 4)
Meteorological measurements and snow depth from an Automated Weather Station.

Manual measurements were performed approximately once a week. The actual measurement days were picked to favour stable
illumination conditions (fully overcast or clear-sky) thus minimising the uncertainties associated with radiative measurements
in changing conditions. All the field sampling and measurements were performed by a single operator for the two seasons,
ensuring a stable protocol detailed in the following section.



## 2.1 Snow spectral albedo measurements

Spectral albedo measurements were acquired using two techniques. First, spectral albedo was acquired automatically every 12 minutes from the 18 February 2017 to 21 May 2018 with the Autosolexs instrument (Picard et al., 2016), installed on the weather station. Autosolexs is a spectral albedometer consisting of two measurement heads at the end of a 3-meter metallic arm

extending out from the weather station structure. The first head is equipped with two cosine light collectors, looking downward and upward. This head measures upwelling and downwelling spectral radiation to compute spectral albedo. The second head is equipped with a single upward-looking collector and measures the spectral ratio between diffuse and total incoming illumination – a mandatory quantity for albedo processing. The diffuse radiation is acquired by hiding the direct solar illumination with a thin strip piloted by a sun tracker. The 3 collectors are connected by fibre optics to an optical switch, itself connected to a

spectrometer. This device acquires one spectral albedo spectrum and one diffuse-to-total illumination spectral ratio every 12 minutes with an effective resolution of 3 nm from 350 to 1050 nm. A full description of this instrument detailing the hardware specifications and data processing can be found in Picard et al. (2016).

Second, on each observation day, at least three different spectral albedo measurements were acquired manually using a single-channel manual version of the Autosolexs instrument (Picard et al., 2016). This albedometer (Solalb; Larue et al.,

2019) is a hand-held instrument made of a single cosine light collector fixed at the extremity of a 3-meter metal bar, with the same characteristics as Autosolexs. The collector is directly connected to the spectrometer by a fibre optic. To obtain an albedo measurement, an upward-looking measurement is first acquired followed by a downward-looking measurement. This operation usually takes up to 30 seconds to execute, the time during which the variations of total incoming irradiance are continuously measured by a photodiode. When the incoming irradiance varied by more than 1% between the upward

and downward acquisitions, the acquisition was discarded. A digital inclinometer located on the measurement head gives an instantaneous control on the levelling of the sensor, with a 0.1 degree accuracy. The manual measurements were performed at the same position as the snowpit physical measurements, before any surface disturbance. After the albedo acquisition, the diffuse-to-total ratio is measured by hiding the sun with a thin strip, as with Autosolexs except that the operation is manual. Slope inclination and azimuth of the snow surface under the sensor – that have to be accounted for in the data processing (Dumont et al.,

2017) – are measured after the acquisition. To do so, the azimuth of the greatest slope was first visually determined. Successive measurements of the maximum inclination around this azimuth were then performed to find the maximum inclination.

## 2.2 Snowpit measurements

Snowpit measurements were performed on each field day for the uppermost 20 centimetres of the snowpack at least . For each new measurement, the snowpits were dug at least one meter away from the former snowpit's location, at a distance of 5 to 20

meters from the Automated Weather Station. Tuzet et al. (2019) provide a detailed description of these measurements and the most important features of the dataset are recalled below.

Vertical profiles of snow density and snow specific surface area (SSA) were collected at the same position as the manual albedo acquisitions. Density was measured with a 6 cm vertical resolution using a cylindrical cutter with a volume of 0.5





L. During the snow season 2016-2017, SSA vertical profiles were collected with the DUFISSS instrument (DUal Frequency Integrating Sphere for Snow SSA measurement; Gallet et al., 2009) at a 3 cm vertical resolution, excluding ice layers. During the snow season 2017–2018, SSA profiles were acquired with the Alpine Snowpack Specific Surface Area Profiler (ASSSAP; Libois et al., 2014). The uncertainty associated with these SSA measurements was estimated to be 10% (Arnaud et al., 2011;

Gallet et al., 2009). For both seasons, one to five samples of the surface were measured. For days when several surface samples were collected, the variability was lower than 15% for 95% of the samples, accounting for both the field variability and the measurement uncertainties.

Vertical profiles of dust and refractory Black Carbon (rBC) concentrations were obtained with a 3 cm vertical resolution. For this, snow was sampled in triplicates in sterile 50 mL polypropylene centrifugation tubes. The samples remained frozen until

analysis in the laboratory, where they were melted and immediately analysed after nebulisation, using a Single Particle Soot Photometer (SP$^2$, Droplet Measurement Technologies), according to the procedure described in Wendl et al. (2014). Dust size distributions and concentrations were measured with a Coulter Counter following Delmonte et al. (2004). The measured sizes span a range of 0.6 to 21 $\mu$m and we assume here that insoluble particles above 0.6 μm are mainly dust particles.

The dataset also includes Organic Carbon (OC) and Elemental Carbon (EC) measurements in addition to the dataset described

in Tuzet et al. (2019). Snow was sampled with a vertical resolution of 10 to 20 cm, following Voisin et al. (2012), with stainless steel instruments and stored frozen (-30°C) in pre-cleaned borosilicate glass bottles until further processing. Snow was then melted, and a coagulant ($NH_4H_2PO_4$, 1.5g / 100mL; Torres et al., 2014) was added before filtering on precombusted QMA filters. The addition of the coagulant lets BC particles agglomerate and helps increase the filtration efficiency for BC to $\approx$ 90%. This is a key step, as QMA quartz filters in water have typical cutoff diameters around 500nm (Lim et al., 2014; Torres

et al., 2014). EC/OC was then quantified on the entire filter (21 mm) by a Thermal Optical Transmission method (Sunset Lab instrument) following the EUSAAR-2 protocol (Cavalli et al., 2010).

In this study, we focus on near-surface SSA and LAP concentrations. To obtain these near-surface properties, all the samples of SSA, dust, rBC and EC collected in the top layer are averaged. Note that the EC dataset has a lower vertical resolution (10 or 20 cm) inducing higher uncertainties in the near-surface concentrations.

## 2.3   Ensemble snowpack simulations

Two ensemble simulations of the Crocus snowpack model (Lafaysse et al., 2017) were performed at an hourly time step to simulate the evolution of the snowpack at our study site with and without LAPs. The meteorological forcing and deposition fluxes, as well as the model configuration, are detailed in the following sections.

### 2.3.1   Meteorological forcing

During the two snow seasons, the weather station installed on the study site recorded most of the variables needed to run Crocus, namely air temperature, shortwave and longwave incident radiation, wind speed, atmospheric pressure and relative humidity. For each variable, the forcing for a time step t is computed as the mean of available measurements between t-1h and t. Concerning precipitations, the timing of the precipitation events is taken from SAFRAN reanalysis (Durand et al., 1993) for 2016–2017





and from an unshielded rain gauge installed at the study site for 2017–2018. For both years, the intensity of the precipitations is manually adjusted to reproduce snow depth variations and the phase of the precipitation is determined with a threshold at +1°C (e.g. Aðalgeirsdóttir et al., 2006). Indeed, SAFRAN provides information covering a 1000 km$^2$ area which can not reflect exactly the local precipitation amount. Furthermore, the intensity of the precipitation measured with the rain gauge – installed

just before the beginning of 2017–2018 snow season– was underestimated even after state-of-the art corrections (Klok et al., 2001, Kochendorfer et al., 2017, Morin et al., 2012; data not shown here).

### 2.3.2 Forcing of LAP deposition fluxes

Aerosol deposition fields come from the regional climate model ALADIN-Climate version 6, described by Daniel et al. (2019). This model includes an interactive tropospheric aerosol scheme, named TACTIC (Tropospheric Aerosols for ClimaTe In CNRM)

and presented in Nabat et al. (2015) and Drugé et al. (2019). TACTIC is able to represent the main anthropogenic (sulfate, BC, organic matter, nitrate and ammonium) and natural (dust and sea-salt) aerosol species in the troposphere. These 7 aerosol types are represented through 16 sectional bins also including two gaseous precursors (sulfur dioxide and ammonia), which are prognostic variables in the model: i.e. subject to transport (semi-lagrangian advection, and convective transport), dry deposition, in-cloud and below-cloud scavenging. A simulation was carried out over a regional domain covering Europe, the Mediterranean

Sea and Northern Africa, at a 12 km horizontal resolution with 91 vertical levels. From this simulation, the hourly outputs of BC and dust deposition were extracted at the closest grid point to the Col du Lautaret. The deposition fields include both dry and wet deposition mechanisms and are then given as input to the Crocus model to compute the evolution of LAPs and the radiative impact on the snowpack (Tuzet et al., 2017).

### 2.3.3 Simulation framework

Ensemble simulations are performed with the multiphysics version of Crocus model (ESCROC; Lafaysse et al., 2017). ESCROC is an ensemble framework running several Crocus simulations with different configurations of snow physical processes to represent modelling uncertainties; each simulation of an ESCROC experiment is called a member. Each member is run using the forcing data presented in the previous sections assuming flat terrain. The height of wind and temperature forcing in the simulations is adjusted to the weather station configuration , i.e. 5.18 m and 3.53 m from the ground respectively. The sensors'

distance from the snow surface is then computed by Crocus using the simulated snow depth along the season similarly to the Crocus experiment in Krinner et al. (2018). Moreover, LAP scavenging by meltwater was disabled here because Crocus percolation schemes are highly uncertain. This means that LAPs can not be transported downward in the snowpack by meltwater.

The ensemble used here is composed of 35 members and is similar to the Ensemble E2 described in Lafaysse et al. (2017), whose dispersion has been optimised for a mid-altitude alpine site located less than 100 km from our study site. The only

difference with E2 is that TARTES radiative transfer scheme is used for all the members of our study, a requirement of the LAP implementation (Tuzet et al., 2017). The configuration of TARTES is the same for all members of the ensemble and is described in Section 3.1. As a consequence, the modelling uncertainties on radiative transfer scheme are not accounted for.





To investigate the impact of LAP on snowpack evolution, two ensemble simulations are performed. The first one is forced by ALADIN-Climate LAP deposition fluxes and is referred to as the LAP simulation. The second one has the same 35 members but does not account for the impact of LAPs and is referred to as the pristine simulation. The comparison of the LAP simulation with the pristine simulation provides a numerical estimation of LAP impact on snow evolution as well as the associated modelling uncertainties.

## 3   Methods

### 3.1   Radiative transfer modelling

In the present study, the radiative transfer modelling is based on AART theory (Kokhanovsky and Zege, 2004) as formulated in TARTES. The following assumptions are used throughout the manuscript:

- The refractive index of ice is taken from Warren and Brandt (2008).

- We assume that LAP absorption is only due to BC and dust.

- The MAE (mass absorption efficiency) of the LAPs is considered as known. For BC, MAE is derived from the constant BC refractive index advised by Bond and Bergstrom (2006) i.e $m = 1.91 - 0.79i$; the relation is scaled to obtain a MAE value at 550 nm of 11.25 m$^2$ g$^{-1}$ (Hadley and Kirchstetter, 2012). The scaling makes it possible to implicitly account for the potential absorption enhancement due to internal particle mixing or particle coating. For dust, as the Saharan desert is the major source of particles observed in European Alps (e.g. Thevenon et al., 2009; Di Mauro et al., 2019), MAE is set to the value for dust from Libya with a diameter inferior to 2.5 $\mu$m (PM 2.5) suggested by Caponi et al. (2017). This MAE value was chosen amongst several values reported for Saharan dust, because of its good agreement with our spectral albedo measurements. The impacts of this choice are discussed in Tuzet et al. (2019).

- The shape parameters B and g, used to describe the impact of the ice matrix shape in AART theory, are constant over time and the same for all types of snow. These parameters have a small dependency to the wavelength which is a function of the real part of ice refractive index $r_i$ (taken here from Warren and Brandt, 2008) and are expressed as:

$$B(\lambda) = B_0 + 0.4(r_i(\lambda) - 1.3) \tag{1}$$

$$g(\lambda) = g_0 - 0.38(r_i(\lambda) - 1.3). \tag{2}$$



This implementation is adapted from Appendix F of Libois (2014) and comes from Kokhanovsky and Zege (2004)'s theory. The enhancement parameter $B_0$ is set to 1.6 and the asymmetry factor $g_0$ is set to 0.845 consistently with previous studies (Libois et al., 2014; Dumont et al., 2017).

### 3.2 LAP and SSA estimation from spectral albedo

Snow spectral albedo varies with snow SSA and LAP concentrations (e.g, Warren and Wiscombe, 1980) as well as with the solar zenith angle and the relative proportion of direct and diffuse radiation. To present a comprehensive interpretation of spectral albedo measurements, the spectra are processed to retrieve near-surface SSA and LAP Absorption Equivalent Concentrations (AECs). The AEC is defined as the concentration of LAPs that would cause the same decrease of the visible albedo under the assumption of MAE detailed in Section 3.1 To this end, a 3-step method based on Dumont et al. (2017) is applied. This method
consists in optimising the simulated spectral albedo – accounting for direct and diffuse incoming radiation as well as the slope of the surface under the sensor– to estimate SSA and AEC as follows:

1. A scaling value (A), accounting for the small errors of cross-calibration between the upward and downward-looking sensors is estimated over a full season using acquisitions during fully cloudy days when the irradiance is 100% diffuse.

2. Optimal values of AEC and SSA are estimated from the spectrum between 400 and 1050 nm accounting for slope, aspect
and solar zenith angle. Dumont et al. (2017) consider BC as the only LAP in their AEC estimation, explicitly mentioning the discrepancies caused by this assumption when dust is present near the surface. To overcome this issue, the formulation of the absorption coefficient $\sigma$ (Equation (5) of Dumont et al. 2017) is modified to explicitly account for dust absorption as follows:

$$\sigma(\lambda, \text{SSA}, c_{BC}, c_{dust}) = \sqrt{\frac{32}{3\text{SSA}(1-g(\lambda))} \times \left(\frac{4\pi n_{ice}(\lambda)B(\lambda)}{\lambda\rho_{ice}} + c_{BC}\text{MAE}_{BC}(\lambda) + c_{dust}\text{MAE}_{dust}(\lambda)\right)}. \qquad (3)$$

where SSA is the snow specific surface area, $\lambda$ is the wavelength. $n_{ice}$ and $\rho_{ice}$ are the density and the imaginary part of ice refractive index respectively. $c_i$ and $\text{MAE}_i$ are the AEC and the mass absorption efficiency (MAE) of the impurity type $i$. The optimal concentration of BC and dust are then determined under the assumptions of MAE described in Section 3.1.

3. Each day, the slope and aspect of the snow surface under the sensor are estimated from the diurnal cycle of spectral albedo.

This method is applied to the albedo acquisitions during which the solar zenith angle was lower than 65° to avoid uncertainties
arising with low illumination angles. The estimations of near-surface LAP concentrations and SSA for which the root mean squared error (RMSE) between the optimal spectrum and the albedo measurement is higher than 0.022 are discarded. More details about the method can be found in Dumont et al. (2017) and additional details about the Autosolexs retrievals are presented in Appendix A. A similar method is also applied to manual spectral albedo to retrieve near-surface SSA and AEC. In this case,





we use the slope measured manually as an input of the retrieval algorithm, since the slope estimation can only be computed when a diurnal cycle of albedo is measured. At last, the same method is applied to the spectral albedo computed by Crocus/TARTES (Tuzet et al., 2017) except that the terrain is considered to be flat. The retrieval is executed for each member of the LAP and pristine simulations at noon every day when there is snow on the ground. This provides an ensemble of near-surface snow SSA

and near-surface AEC.

### 3.3    LAP concentration terminology

This study aims at comparing chemically measured LAP concentrations with absorption equivalent concentrations (AEC) estimated from spectral albedo measurements. Here, we chose to present all concentrations as equivalent BC concentrations (eqBC; e.g. Dumont et al., 2017; Tuzet et al., 2019). This eqBC concentration represents the amount of BC that would induce

the same absorption as both dust and BC actually present in the snowpack.

For both measured LAP concentrations and optically estimated AECs, the eqBC concentration is calculated as:

$$c_{\mathrm{eqBC}} = c_{\mathrm{BC}} + \psi(c_{\mathrm{dust}}),\tag{4}$$

where $c_{\mathrm{BC}}$, $c_{\mathrm{dust}}$ and $c_{\mathrm{eqBC}}$ are the BC, dust and eqBC concentrations respectively. $\psi$ is a function computing the BC concentration that would have the same integrated radiative impact from 350 to 900 nm as the input dust concentration. More

details about the computation of $c_{\mathrm{eqBC}}$ with the same values of BC and dust MAE are given in Tuzet et al. (2019).

In the following, eqBC concentrations from chemical measurements are referred to as eqEC and eqrBC concentrations and are computed applying Equation 4 with $c_{\mathrm{BC}}$ equal to the measured rBC or EC respectively.

The fraction of total LAP absorption which is caused by dust ($\eta$) is computed as follows:

$$\eta = \frac{\psi(c_{\mathrm{dust}})}{c_{\mathrm{eqBC}}},\tag{5}$$

with $\eta = 1$ indicating that the radiative impact is solely caused by dust and $\eta = 0$, by BC.

### 3.4    Handling of the ensemble simulation

Most of the simulation results are represented by the median and the spread (minimum and maximum values) of the ensemble. Due to the ensemble size of 35 members, this spread can be interpreted as the 95% confidence interval of a given diagnostic. Unless otherwise specified the spread and the median of the ensemble are calculated considering only the members with snow

on the ground. For each member with snow on the ground, several quantities are computed to represent the impact of LAPs and are listed hereafter. Note that all statistics presented in the manuscript exclude periods when the measured automatic snow depth was lower than 20 cm to ensure that albedo measurements are not affected by the signal of the ground.



**Radiative forcing (RF)**

At each time step, the instantaneous radiative forcing (RF) of LAP is computed as the difference between the energy absorbed by the whole snowpack in the LAP and the pristine simulations ($E_{\text{LAP}}$ and $E_{\text{pristine}}$ respectively):

$$RF = E_{\text{LAP}} - E_{\text{pristine}} \ [Wm^{-2}]. \tag{6}$$

The daily RF is computed as a 24 h average of the instantaneous RF.

**Indirect impact**

The indirect impact of LAPs is also estimated at each time step as detailed in Tuzet et al. (2017). A simulation accounting only for the indirect impacts of LAP is performed with TARTES offline. For each time step, a TARTES calculation is made using the snowpack physical properties – i.e. the SSA, thickness and density of each layer – of the LAP simulation, but no LAPs. This

way, the direct impact of LAP is ignored and only the radiative impact due to change of snow metamorphism is accounted for. This simulation is hereafter referred to as the indirect simulation. For each member and at each time step, the indirect impact $R_{ind}$ is computed as

$$R_{ind} = \frac{E_{\text{LAP}} - E_{\text{indirect}}}{E_{\text{LAP}} - E_{\text{pristine}}} = \frac{RF_{\text{indirect}}}{RF}, \tag{7}$$

where $RF_{\text{indirect}}$ is the indirect radiative forcing. The daily $R_{ind}$ is computed as the ratio between the daily RF of the indirect

simulation and the daily RF of the LAP simulation.

**Shortening of the snow season**

For each season, the date of definitive disappearance of the snowpack ($t_{\text{melt-out}}$) is also computed as the last date where there is at least 2 kg m$^{-2}$ water equivalent of snow on the ground. The difference between the melt-out dates in the LAP and pristine simulations is written $\Delta t_{\text{melt-out}}$ and corresponds to the shortening of the snow cover duration induced by LAPs.

**4   Results**

**4.1   Two contrasted snow seasons**

Figure 2 shows the observed evolution of the snow depth and of the meteorological conditions over the snow seasons 2016–2017 and 2017–2018. Figure 2 a) shows the evolution of the snow depth measured by the weather station and the operator on each field day. Rain-on-snow (ROS) events are represented as blue vertical lines and are determined based on the precipitation forcing

(see Section 2.3.1). The brown shading corresponds to a major dust deposition that occurred at the beginning of April 2018. Manual and automated measurements are in good agreement, with snow depth differences below 30 cm, despite the horizontal distance between the automatic sensor and the manual measurements (up to 20 m). This highlights the moderate spatial snow



depth variability within the study area, despite the occurrence of frequent wind events with wind speeds higher than 6 m s$^{-1}$ for both years (Figure 2 c).

The evolution of the snowpack is different between the two years. 2016–2017 features low accumulation, with few snowfall events in December and January and most of the snowfalls mixed with rain in February and March. The ablation phase started
early, around mid-March and the snowpack first disappeared around 20 April 2017. Two small snowfalls in May rebuilt an ephemeral snowpack lasting around two weeks. In 2017–2018, the accumulation was higher with many snowfall occurrences from December to mid-April. The ablation phase started approximately one month later, around mid-April and the snowpack completely disappeared one month later around 20 May 2018. Part of these differences can be explained by the temperatures (Figure 2 b) which were higher for the first year especially in March and at the beginning of April. This meteorological overview
underlines the contrast between both seasons in term of snow accumulation, temperatures as well as extreme dust deposition events.

## 4.2   Measured near-surface properties

Figures 3 a) and b) show the evolution of near-surface properties measured with the different methods presented in Sections 2 and 3. The near-surface AEC and SSA retrieved from Autosolexs correspond to the daily median value and the error bars
correspond to the first and third quartiles of all valid daily measurements. The evolution of these near-surface properties is related to snowfall events and melt phases as illustrated with the evolution of the snow depth in Figure 3 c).

Figure 3 a) presents the evolution of near-surface LAP concentrations. In general, a decrease of near-surface concentrations is observed after snowfalls whereas an increase is observed during the melt periods. This surface enrichment is particularly marked at the end of the two snow seasons as the snowpack undergoes strong melt, and LAPs of the melting layers accumulate at the
surface (e.g. Sterle et al., 2013). In the second year, a major dust deposition occurred at the beginning of April (brown shading) and was immediately buried by new snowfalls until it reappeared at the surface mid-April, contributing to the observed high LAP concentrations. Table 1 presents the RMSE, bias and Pearson correlation coefficients (r$^2$) between different estimates of near-surface LAPs, highlighting several conclusions:

1) Both eqrBC (rBC+dust) and eqEC (EC+dust) concentrations from chemical measurements show a good correlation (r$^2$
$\simeq$ 0.86) but eqEC is almost systematically higher than eqrBC. To better understand this bias, a direct comparison of EC and rBC measurements for all available samples, including samples that are not near the surface, is presented in Figure 4. High discrepancies between both BC measurement techniques can be noted with a ratio EC/rBC ranging from 0.5 to 30, and a mean value around 10.

2) The AECs retrieved from Autosolexs shows an equally good correlation with the LAP concentrations from both types of
chemical measurements (r$^2$ $\approx$ 0.75). Autosolexs AECs are systematically higher than chemical measurements, meaning that the measured concentrations are too low to explain the impact on albedo with our assumed MAE values. The difference is more pronounced for rBC measurements than for EC measurements in line with the results of Figure 4. The causes of these discrepancies are further discussed in Section 5.1. The correlation between Solalb AECs and chemical measurements is low (r$^2$ $\approx$ 0.3 and 0.44 for eqEC and eqrBC respectively). This is mainly due to three measurement days during the first season when





Solalb AEC values are higher than 100 ng g$^{-1}$ eqBC while chemical measurements are lower than 50 g$^{-1}$ eqBC. The cause of these three outliers is unknown. The values in blue in Table 1 correspond to the statistics computed without these outliers. Once these points are removed, the correlation is good (r$^2$ ≈ 0.88 and 0.72 with eqrBC and eqEC respectively). EqrBC concentrations are too low to explain Solalb AEC similarly to Autosolexs AECs. However, there is no significant bias between Solalb AECs

and eqEC measurements. Two of the outlier days are before the beginning of Autosolexs measurements, which may explain why the correlation between Autosolexs and chemical measurements is not deteriorated.

Figure 3 b) shows the evolution of measured and retrieved near-surface SSA. In the case of measurements, higher values are generally observed during the second snow season compared to the first one. High SSA values are usually observed for fresh and cold snow (Legagneux et al., 2002), that was rarely present at the surface during the 2016–2017 season owing to the warm

and wet meteorological conditions of the season. The measured SSA correlates better with SSA retrieved from manual albedo (r$^2$ = 0.82) than with the one retrieved from automated albedo measurements (r$^2$ = 0.58). This may be explained by manual albedo measurements and SSA measurements being performed at the same place and most of the time in a 1 hour time interval, whereas automated albedos are collected up to 20 m from the SSA measurements and are represented by the daily median.

### 4.3   Ensemble simulations

Figures 5 a) and b) show respectively the near-surface AEC and SSA simulated at noon compared to field measurements. For sake of clarity, simulations were only compared to a single source of measurement: the automated spectral albedos from Autosolexs. We chose Autosolexs because 1) it measures the quantity of interest for this study: the radiative impact of LAP and 2) it has a higher temporal resolution than the manual measurements. All the statistics presented in this section do not account for values in the grey-shaded area, for which the measured automatic snow depth is lower than 20 cm to ensure that Autosolexs

measurements are not influenced by the ground.

Figure 5 a) shows the evolution of near-surface AECs estimated from Autosolexs and simulated by the LAP ensemble. The simulated AEC median is correlated with the AEC retrieved from Autosolexs (r$^2$ ≃ 0.78), meaning that temporal variations of near-surface AEC are correctly reproduced. The simulated LAP absorption is lower than that estimated from Autosolexs, with an AEC bias of around 31 ng g$^{-1}$. This means that the median LAP absorption in our simulation is slightly underestimated by

our modelling framework. The agreement between the ensemble simulations is lower in April 2018, just after the strong Saharan dust deposition. The dispersion of near-surface LAP concentrations in the ensemble is quite low regarding the median value, which is not surprising because the LAP deposition fluxes and their evolution laws within the snowpack are not perturbed.

Figure 5 b) shows the evolution of measured and simulated near-surface SSA. The temporal patterns of SSA evolution are well captured by the model. For SSA lower than 15 m$^2$ kg$^{-1}$ there is no significant bias between Crocus SSA and the measurements.

However, for higher SSA values there is a clear bias between Crocus and measurements, with Crocus systematically predicting lower SSA values. This bias may be explained by Crocus' parameterisation of fresh snow SSA, which is set to never exceed 65 m$^2$ kg$^{-1}$ whereas values up to 105 m$^2$ kg$^{-1}$ were measured in the field. It is noteworthy that there are no significant SSA differences between the simulations with and without LAPs. Finally, the dispersion of near-surface SSA within the ensemble is significant for low SSA values, which sometimes vary from 5 to 20 m$^2$ kg$^{-1}$ depending on the member. This dispersion





is explained by the 3 different metamorphism laws used in ESCROC together with the indirect effects of the other perturbed processes.

Figure 5 c) shows the evolution of simulated and measured snow depth over the two snow seasons. There is no significant impact of LAPs on snow depth evolution before the beginning of the ablation phase –i.e, 6 March for the first year and 13 April

for the second one. Over this period, both the pristine and the LAP simulations reproduce well the measured snow depth with an RMSE of $\approx 6.5$ cm. This is expected as the precipitation forcing was adjusted to fit the snow depth data as used here for the evaluation. Nevertheless, the snow depth observed in the measurements during the ablation phase is better reproduced by the LAP simulation than by the pristine simulation (RMSE $\approx 5.8$ cm and 17.2 cm for the median of LAP and pristine simulation respectively on this period), because the melt-rate is underestimated by the pristine simulation. The dispersion of the ensemble

is high in both simulations, with snow depth varying by up to 0.8 m depending on the member.

### 4.4   LAP radiative impacts and consequence on melt

For both snow seasons, the highest AECs estimated from Autosolexs are within the simulated concentrations by our ensemble, meaning that the extreme values of the simulated RF of LAPs are expected to be representative of Autosolexs measurements.

Figure 6 a) shows the daily RF in W m$^{-2}$ of LAP estimated from ensemble simulations, which increases with time during

each snow season as more shortwave energy becomes available from winter to spring. It becomes particularly important during the final ablation phase (Figure 6 c), with the enrichment of LAPs at the surface of the snowpack. This trend is modulated by snowfalls that lead to lower surface eqBC concentrations and snow cover that lowers the amount of incoming radiation. The seasonal RF of LAPs, computed as the sum of all daily RF, is 1.33 times higher for the second season than for the first one. This is due both to the later triggering of the melt-out phase (and hence occurring with higher solar radiation) and to the higher eqBC

concentration (Figure 5 a). The maximum daily RF values of LAPs are estimated around 38.2 and 55.8 W m$^{-2}$ for the first and the second year respectively. Hourly values of the RF (not shown) peak at 125 and 215 W m$^{-2}$ for the first and the second year respectively.

Figure 6 b) shows the evolution of the fraction of the RF of LAPs coming from the indirect impact. The lightest envelope corresponds to all the members while the darkest one corresponds to the first and third quartiles to disregard the impact of

outliers. The indirect impact of LAPs strongly varies with time and can be either positive or negative. For both snow seasons, the indirect impact cumulated over the whole season is close to zero (-1 and +1 % of the total RF for 2016–2017 and 2017–2018 seasons respectively). This result is further discussed in Section 5.2.2.

Figure 7 shows the evolution $\Delta t_{\text{melt-out}}$ – i.e. the number of days by which the snow season is shortened – as a function of the melt-out date of the pristine simulation expressed in Day Of Year (DOY). Each point corresponds to a member of the ensemble

simulations. The median $\Delta t_{\text{melt-out}}$ is represented by the horizontal lines for each season. $\Delta t_{\text{melt-out}}$ is respectively 10 and 11 days for the first and the second snow season. For the first snow season, $\Delta t_{\text{melt-out}}$ varies widely with the model configuration with most values ranging from 7 to 20 days. For the second snow season, $\Delta t_{\text{melt-out}}$ exhibit a small dependency to the model configuration, with most values between 8 and 12 days.





### 4.5 Apportionment between BC and dust impacts

Understanding the distribution of the direct RF between dust and BC is of importance to model LAP impacts (Skiles et al., 2018), especially at the end of the snow season when this RF is maximal. For both seasons, the RF of LAPs remains low (< 5 W m$^{-2}$ daily RF) until the final ablation phase – occurring after the peak of snow accumulation for both seasons (Figure 6 a).

Figure 8 a) shows the evolution of $\eta$, i.e. the proportion of the RF of LAPs attributed to dust during the final ablation phase.

In the simulations, less impact is attributed to dust for the first year (around 35% in median) than for the second one (around 55% in median). This is mainly due to the major dust deposition of the beginning of April 2018 that outcrops on 19 April, resulting in a high $\eta$ value of around 0.85. A clear conclusion is hard to draw for the measurements because of the strong discrepancies between the different estimates of $\eta$. First, Autosolexs retrieval attributes the whole impact either to BC or to

dust, with few intermediate $\eta$ values. The only information detected is whether dust or BC dominates LAP absorption with no precise quantification of $\eta$. Second, the values of $\eta$ estimated from chemical measurements are higher for eqrBC than for eqEC, due to the strong discrepancy between both BC measurements (Figure 4). The $\eta$ values corresponding to eqEC are generally closer to simulated values rather than when obtained from rBC values. Considering all measurements together, it seems that dust contributes more to the impact of LAPs during the second season than during the first one, which is in agreement with Crocus

simulations. However, simulated $\eta$ are almost systematically lower than any measurement over both seasons, meaning that too much impact is attributed to BC in the simulation.

## 5 Discussion

The previous section presents a comparison between different measurements of near-surface SSA and near-surface LAP concentrations, highlighting large discrepancies between EC and rBC concentration measurements and pointing out an issue in

our understanding of the related processes. This issue is of particular importance to link chemical concentrations of LAPs to their absorption as discussed in Section 5.1. As MAE uncertainties are already extensively discussed in Tuzet et al. (2019), the focus is placed here on the chemical measurements. The measurements are then compared to ensemble snowpack simulations performed with Crocus. The modelled near-surface AECs are in good agreement with automated albedo observations, meaning that LAP absorption is correctly simulated. Despite the strong difference of RF between the two seasons, the median $\Delta t_{\text{melt-out}}$

are similar for both seasons (10 and 11 days respectively; Section 5.2.1). Finally, there is no significant indirect impact of LAP these to snow seasons, contrary to what was observed in previous studies (Section 5.2.2).

### 5.1 On chemical measurements in snow

Figure 3 points out a marked bias between eqrBC concentrations and AEC retrieved from spectral albedo measurements, the chemical concentrations being lower. This suggests that the eqrBC concentrations measured chemically in the snowpack are too

low to explain the observed LAP absorption under state-of-the-art assumptions about BC MAE. These findings confirm those of Doherty et al. (2016) and Tuzet et al. (2019).



From atmospheric science, we know that MAE values determined from experiments strongly depend on both concentration and absorbance measurement techniques (e.g. Salako et al., 2012; Chan et al., 2011, 2010; Zanatta et al., 2016; Ram and Sarin, 2009; Venkatachari et al., 2006). For instance Chan et al. (2010) suggest that the MAE of BC ranges from 2 to 6 m$^2$ g$^{-1}$ when determined from EC concentration against 8 to 55 m$^2$ g$^{-1}$ when determined from rBC concentrations (Chan et al., 2011).

Watson (2005) indicates similar differences of about 1 order of magnitude.

In our case, the observed bias between AEC from spectral albedo and chemistry is significantly reduced when using eqEC measurements instead of eqrBC (Figure 3 a). This is not surprising given the significant bias between eqrBC and eqEC concentrations (Table 1). When focusing on the two types of BC measurements, it appears that for all valid snow samples of our dataset, EC measurements are 10 times higher than rBC measurements on average but EC/rBC ratios range from 0.5 to 30

depending on the sample (Figure 4). Similar findings have recently been observed for Arctic snow in the dataset of Mori et al. (2019) and are not unexpected.

This suggests that the value of BC MAE can not be chosen independently of the measurement technique used for BC. The value used in the present study (11.25 m$^2$ g$^{-1}$ at 550 nm) is well adapted for EC but too low for rBC. Similar issues are likely to affect dust measurements, which could explain the divergence between optically estimated AEC and both chemical

concentrations at the end of 2017-2018 snow season (when all the estimates of $\eta$ vary from 0.9 to 1). This issue comes on top of the strong variability reported for LAP absorption efficiency and makes it very challenging to link LAP chemical concentrations to their radiative impact.

## 5.2    On the impact of LAP on snow cover evolution

### 5.2.1    Variability of $\Delta t_{\text{melt-out}}$

Figure 6 shows the evolution of the RF of LAPs over the two snow seasons. The maximum values of daily and instantaneous RF, around 50 and 200 W m$^{-2}$ respectively, are in range with maximum values for Europe as suggested by Skiles et al. (2018). The seasonal radiative forcing of LAPs during the second snow season is higher than during the first one by a factor 1.33. Nevertheless, the median advance of melt-out date due to LAP is close for both seasons. Surprisingly, the maximum $\Delta t_{\text{melt-out}}$ estimated for the first season (20 days) is much higher than for the second one (12 days). These contradictory facts can be

reconciled by considering the differences of meteorological conditions between the two snow seasons. Indeed, during the first snow season, two small snowfalls occurred at the beginning of May (Figure 2 a), which deposited $\approx$ 40 kg m$^{-2}$ of snow water equivalent according to all the members of the pristine simulation. In contrast, for the LAP simulation, the evolution of these snowfall events strongly varies from a member to another, depending on whether the simulated snowpack has already totally melted out at the beginning of May or not. If the two snowfalls are deposited on bare ground, all or part of the snowfall does

not hold on the ground depending on the member. All $\Delta t_{\text{melt-out}}$ higher than 15 days correspond to members in which both snowfalls immediately melt, leaving the ground bare. On the opposite, for members in which the snowfalls are deposited on top of an existing snowpack in LAP simulations – for which the pristine simulation melts after the DOY 135 (15 May) – the median impact is around 8 days, in better agreement with the difference of RF between both years. The intermediate cases





correspond to a partial melt of the snowfalls deposited on the ground. These different scenarios explain the high variability of $\Delta t_{\text{melt-out}}$ observed for the first snow season. The lower variability of $\Delta t_{\text{melt-out}}$ between the configurations for the second snow season is explained by the continuity of the snow-cover until the end of the season for all members. These findings highlight the necessity to account for the complex interplay between LAP dynamics in the snowpack, meteorological conditions and the

snowpack/ground interactions to accurately estimate LAP impact on melt.

### 5.2.2    Indirect impact

Previous studies have shown that the indirect impact of LAPs – caused by the enhancement of SSA decrease due to LAP direct RF – represents around 20% of LAP total RF (e.g. Tuzet et al., 2017; Skiles and Painter, 2019). Over the two years considered here, there is no clear influence of the indirect impact of LAPs. Integrated over the season, the median portion of the RF of LAPs

coming from the indirect impact are around -1% and +1% for the first and second year respectively. Furthermore, the strong dispersion of the ensemble indicates that the diagnostic is highly sensitive to the parameterisation of other physical processes in the snowpack model. It is hence of particular importance to account for modelling uncertainties to investigate this process. The negative impact, that may be surprising at first, can be explained by the counter-intuitive outcropping of sub-surface layers with a higher SSA than the surface layer in Crocus. Indeed, in some cases, the presence of LAPs accelerates the numerical

outcropping of a snow layer with a high SSA, while this layer remains buried by snow with low SSA in the pristine simulation. This can lead to larger energy absorption by the pristine simulation than by the indirect simulation. Given that this process may be physically plausible, we did not discard negative indirect impact values in this study.

The decrease in SSA induced by LAPs is particularly important when snow has a high SSA and, at the same time, contains a large concentration of LAPs (Tuzet et al., 2017). These conditions were rarely observed over the two seasons considered here

(Figure 5). ROS events and the warm temperatures of the first year maintained low surface SSA values during a major part of the season. During the second season, the only period with high LAP concentrations (> 50 ng g$^{-1}$ eqBC) and high SSA values ($\approx$ 40 m$^2$ kg$^{-1}$) is around 15 March 2018. During the two following weeks, the indirect impact of LAP was around 15–20%. These findings suggest a strong dependence of the indirect impact of LAPs to both meteorological conditions and LAP deposition. Under the conditions observed during the two seasons studied here, the indirect impact of LAP is particularly inefficient.

### 5.2.3    BC vs. dust

Figure 5 a) highlights a strong underestimation of near-surface concentrations of LAPs around 20 April 2018. This period corresponds to the outcropping of a layer containing dust from the major deposition that occurred two weeks before. During the following weeks, the measured concentration remained stable while the simulated concentration continued to rise significantly, finally reaching the measured values. During the same time (20 April to 30 April), the value of $\eta$ –i.e. the proportion of LAP

absorption caused by dust – decreases noticeably (Figure 8 a). This feature suggests that the strong dust deposition at the beginning of April is underestimated by ALADIN-Climate and that the match between measured and simulated concentrations at the end of the season is due to compensation by BC. This confirms the findings of Tuzet et al. (2017) who show that major Saharan dust deposition events are underestimated by ALADIN-Climate. More generally, at the end of both seasons, all $\eta$ values





measured are higher than the simulated ones. This means that the good agreement between measured and simulated AEC is due to compensation between the overestimation of BC deposition and the underestimation of dust deposition, reinforcing the hypothesis of Tuzet et al. (2017). As a consequence, care should be taken to expand the conclusion of this study to other regions.

## 6  Conclusion

This study provides an analysis of a unique dataset collected at Col du Lautaret (2058 m a.s.l., French Alps) site during two snow seasons featuring contrasted meteorological conditions. This dataset comprises automated measurements of spectral albedo and meteorological variables as well as 30 days of manual measurements of spectral albedo, and vertical profiles of snow physical and chemical properties. Spectral albedo measurements are first processed to estimate near-surface SSA and AEC of the snowpack. Then the estimates are compared to the snowpit measurements. Near-surface SSA retrieved from spectral

albedo measurements and measured in the snowpit are overall in good agreement. However, our dataset highlights strong discrepancies between different chemical measurements techniques for BC, with a mean EC/rBC ratio around 10 on average and ranging from 0.5 to 30. These results underline the need to better understand what is precisely measured by each chemical measurement technique in snow and how to relate each type of measurements to their radiative impact. Indeed discrepancies between chemical measurements and radiative retrievals from spectral albedo are reduced by using EC measurement rather

than rBC measurements under our assumption of BC MAE (11.25 m$^2$ g$^{-1}$ at 550 nm). This issue is particularly critical for studies focusing on the impact of BC on Arctic and Antarctic snow because in these regions the BC concentrations are often too low to be detected from reflectance measurements. Further studies should aim at comparing the different methods of LAP concentration measurements in snow with concomitant measurements of the induced absorption, as it has been done for BC in atmospheric sciences. Moreover, Bergmann et al. (2019) have recently shown the presence of microplastics in snow in remote

snow-covered areas. We thus recommend further work to determine if this new type of particle in snow 1) has a significant radiative impact; and 2) affects uncertainties of other LAP concentration measurements.

This dataset is also compared to ensemble snowpack simulations performed with the Crocus detailed snowpack model. Two ensemble simulations are performed, one assuming that no LAPs are deposited on the snowpack and another one using BC and dust deposition fluxes from the ALADIN-Climate atmospheric model. This simulation framework makes it possible to

isolate the impact of LAPs on snowpack evolution while accounting for modelling uncertainties. Near-surface properties of Crocus simulations are in good agreement with measured values, except for a marked SSA bias for high SSA values. This bias is probably due to the parameterisation of the SSA of fresh snow in Crocus. Near-surface LAP concentrations computed by ensemble simulations using ALADIN-Climate as a deposition forcing are in good agreement with automated spectral albedo measurements. The temporal evolution, as well as the extreme values at the end of the season, are correctly simulated.

The radiative impact is hence expected to be captured by our simulations even if extreme dust deposition events seems to be underestimated as hypothesised in Tuzet et al. (2017).

By comparing the pristine simulation to the LAP simulation, the RF of LAPs is estimated, with maximal values of 58 W m$^{-2}$ and 215 W m$^{-2}$ for daily averaged and instantaneous RF respectively, over the two seasons. The RF of LAP is higher for

the second snow season which was affected by a major dust deposition event but the median impact on snow cover duration is similar for both seasons: 10 and 11 days respectively. This is due to complex interactions between meteorological conditions and snow-LAP synergy, especially for the first year when the LAP on the duration of snow cover varies from 5 to 20 days depending on the model configuration. Even though the LAP concentrations are similar in each member of our ensemble, there are strong

differences in terms of snow cover duration due to modelling uncertainties. This highlights the need to better constrain snowpack modelling, by assimilating observations for example. As our ensemble framework accurately represents the evolution of visible (AEC concentrations) and near-infrared (SSA) reflectances, it would provide a reasonable first guess for further assimilation of optical satellite reflectance; at least at the studied location for these two particular seasons. An analysis over a longer period and at a larger scale would be needed to extend our findings.

Lastly, our results show that the indirect impact of LAPs – i.e. the enhancement of snow metamorphism induced by LAPs – is negligible. These findings, for the two particular snow seasons studied here at the Col du Lautaret study site, differ from the results of previous studies estimating this impact to be around 15–20 % of LAP total RF. Here strong LAP concentrations near the surface of the snowpack only occur when SSA is already low, explaining the inefficiency of the indirect impact. This suggests that the seasonal indirect impact depends on meteorological conditions and on the timing of LAP deposition on the snowpack. It

is hence necessary to explicitly account for the coupling between LAP deposition and snowpack evolution to reproduce the spatio-temporal variability of LAP indirect impact. This could not be captured with simple snow modelling approaches.

**Author contributions**

F. Tuzet led the study and was in charge of the field measurement campaign over the two seasons. F. Tuzet along with M. Dumont and G. Picard performed the major part of the data analysis. L. Arnaud and G. Picard developed the Autosolexs and

Solalb instruments and the associated processing library. M. Lamare had a significant contribution to improve the writing of the manuscript. D. Voisin supervised the chemical measurement analysis. M. Lafaysse made it possible to realise ensemble snowpack simulations with ESCROC. P. Nabat provided the ALADIN-Climate simulations. M. Lamare, F. Larue and J. Revuelto had a major contribution in the field data acquisition. All authors contributed to the manuscript.

**Acknowledgements**

CNRM/CEN and IGE are part of Labex OSUG@2020 (investissement d'avenir - ANR10 LABX56). This study was supported by the ANR programs 1-JS56-005-01 MONISNOW and ANR-16-CE01-0006 EBONI; the INSU/LEFE projects BON and ASSURANCE; the Ecole Doctorale SDU2E of Toulouse and the CNES APR grant MYOSOTIS. This research was at least partially supported by Lautaret Garden-UMS 3370(Univ.Grenoble Alpes, CNRS, SAJF, 38000 Grenoble, France), member of AnaEE-France (ANR-11-INBS-0001AnaEE-Services, Investissements d'Avenir frame) and of the eLTER-Europe network

(Univ. Grenoble Alpes, CNRS, LTSER Zone Atelier Alpes, 38000 Grenoble, France). The authors are grateful to LISA and PSI for chemical analysis of snow samples presented in this study.



**Code availability**

The code used to produce the figures and process the data are available from the corresponding authors upon request.

**Data availability**

The dataset will be published on an open-access platform (with a DOI) after the review process.

5 **Competing interests**

The authors declare that they have no conflict of interest.



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

**Tables**





| Parameter | eqEC | | Solalb eqBC | | Autosolexs eqBC | |
|---|---|---|---|---|---|---|
| eqrBC | N<br>RMSE<br>Bias<br>$r^2$ | : 23<br>: 23.3 ng g$^{-1}$ eqBC<br>: 15.9 ng g$^{-1}$ eqBC<br>: 0.86 | N<br>RMSE<br>Bias<br>$r^2$ | : 21 (18; no outliers)<br>: 50.4 (23) ng g$^{-1}$ eqBC<br>: 27.5 (12.8) ng g$^{-1}$ eqBC<br>: 0.44 (0.88) | N<br>RMSE<br>Bias<br>$r^2$ | : 12<br>: 106.7 ng g$^{-1}$ eqBC<br>: 66.7 ng g$^{-1}$ eqBC<br>: 0.76 |
| eqEC | | | N<br>RMSE<br>Bias<br>$r^2$ | : 21 (18; no outliers)<br>: 49.2 (24.7) ng g$^{-1}$ eqBC<br>: 13.2 (-3.15) ng g$^{-1}$ eqBC<br>: 0.3 (0.72) | N<br>RMSE<br>Bias<br>$r^2$ | :12<br>: 96.61 ng g$^{-1}$ eqBC<br>: 54.9 ng g$^{-1}$ eqBC<br>: 0.73 |
| Solalb | | | | | N<br>RMSE<br>Bias<br>$r^2$ | : 12<br>: 71.53 ng g$^{-1}$ eqBC<br>: 26.48 ng g$^{-1}$ eqBC<br>: 0.83 |

**Table 1.** Comparison between the different measurements of eqBC concentrations. Each cell of the table contains the number of samples (N) used for the statistics, the RMSE, the bias and $r^2$ between the quantities named in the first column and row. The bias is computed as a difference between the quantity named in the column header and the quantity named in the row header. The values in blue correspond to the statistics after removing the 3 outliers of the solalb analysis.

**Figures**





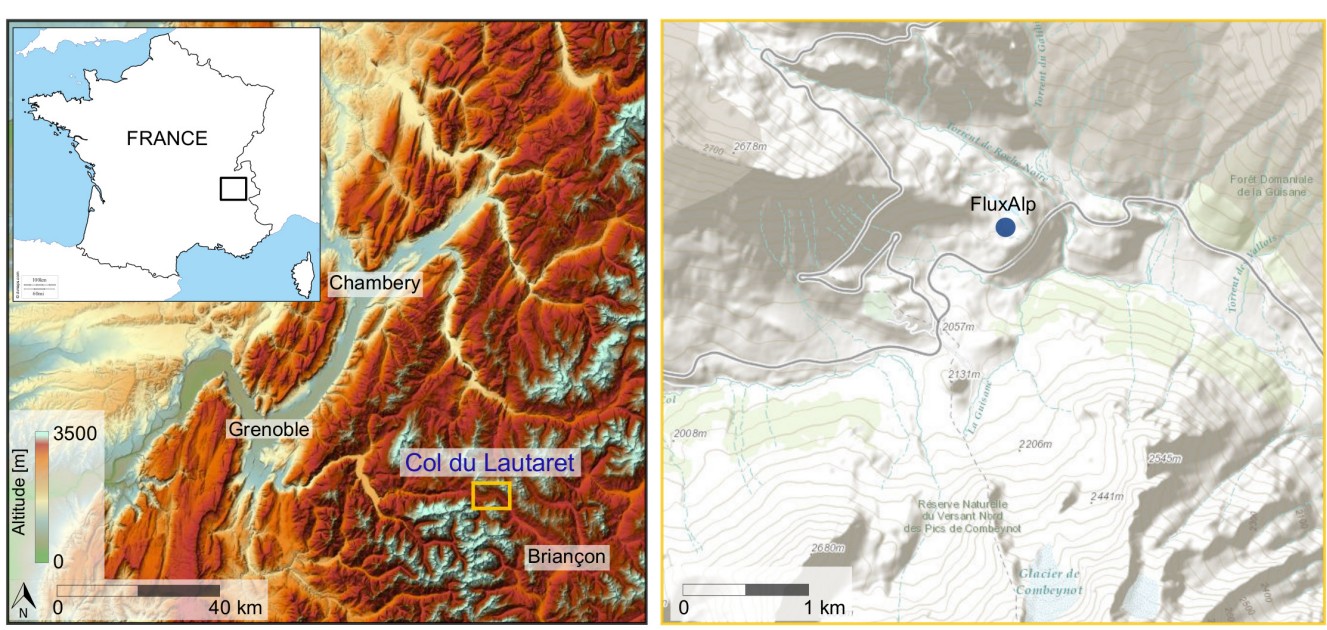

**Figure 1.** Localisation of the study area (left: Col du Lautaret site shown on a hillshade product draped over the 25m DEM (BD ALTI) produced by the French National Institute of Geographic and Forest Information (IGN); right: automated weather station shown on the OpenStreetMap product, © OpenStreetMap contributors 2019. Distributed under a Creative Commons BY-SA License.).



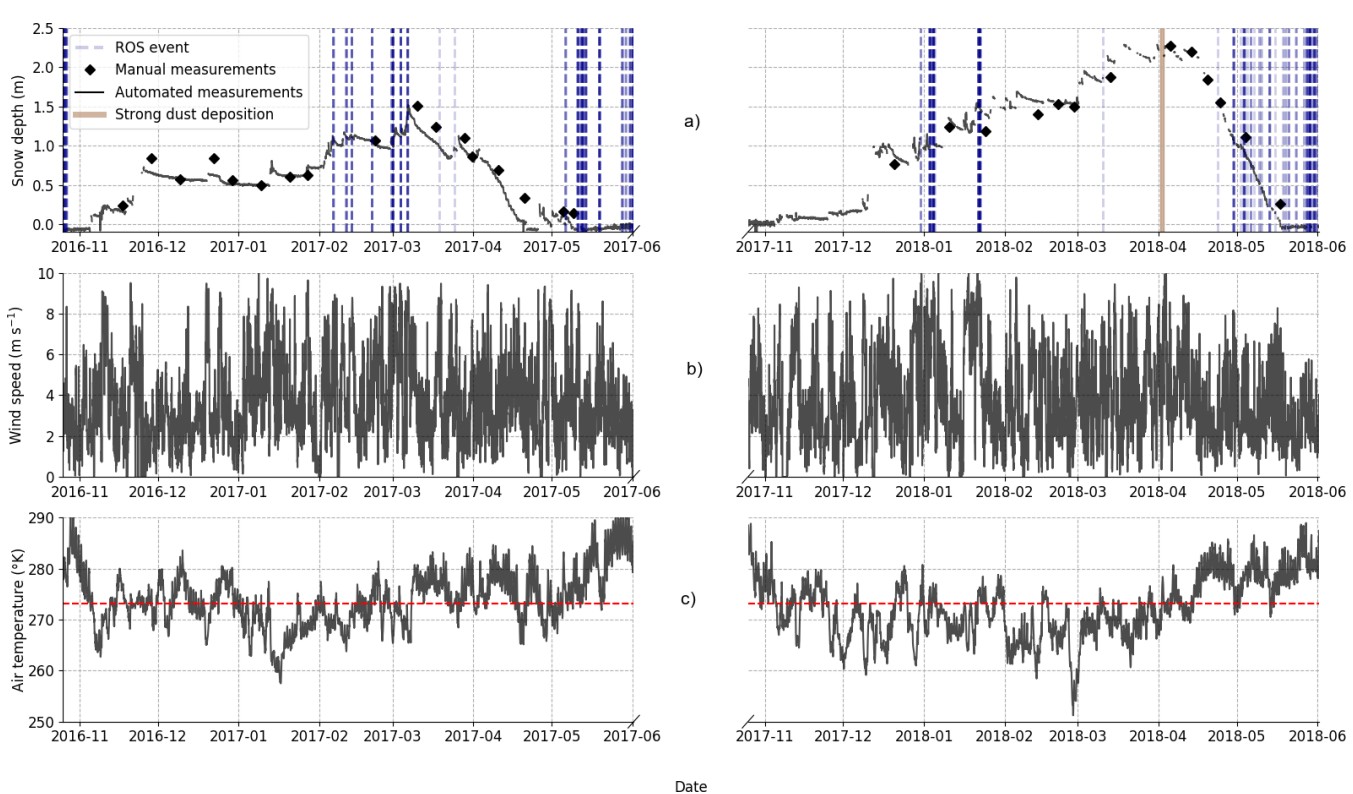

**Figure 2.** Snow depth (a), air temperature (b) and wind speed (c) measured at the weather station for the two snow seasons (black curve). All data are averaged at an hourly time-step. On the upper panel, ROS and strong dust deposition events are represented by dashed blue and brown shadowing respectively. Manual snow depth measurements acquired on each field day are also represented by black diamonds.

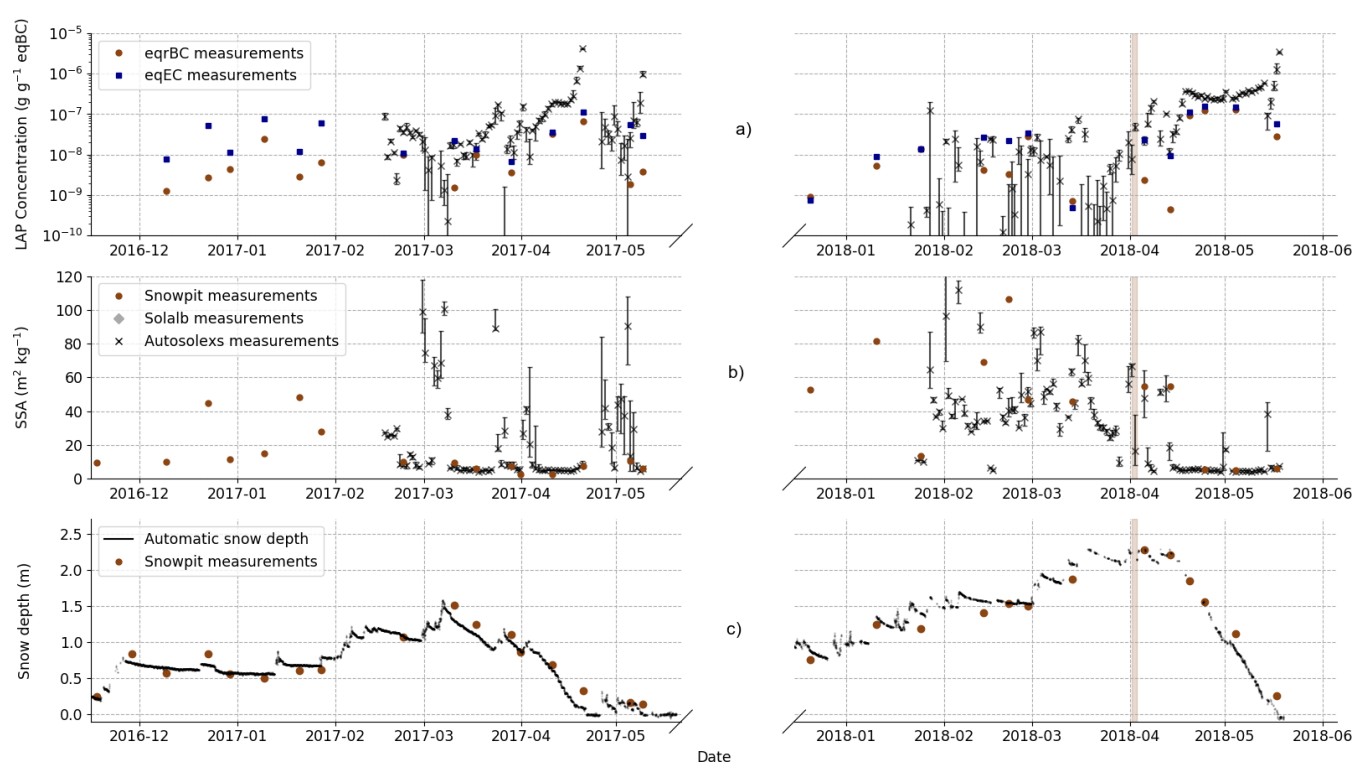

**Figure 3.** Evolution of measured near-surface LAP concentration (a), near-surface SSA (b) and snow depth (c). Information retrieved from automatic spectral albedo (Autosolexs) are represented by black crosses with error bars corresponding to the 25 and 75% quantile of all the measurements of the day. Information retrieved from manual spectral albedo (Solalb) are represented by grey diamonds. Snowpits SSA and eqrBC concentrations are represented by brown dot and eqEC concentrations are represented by blue squares. The major dust deposition event of the second year is represented by the vertical brown shading.



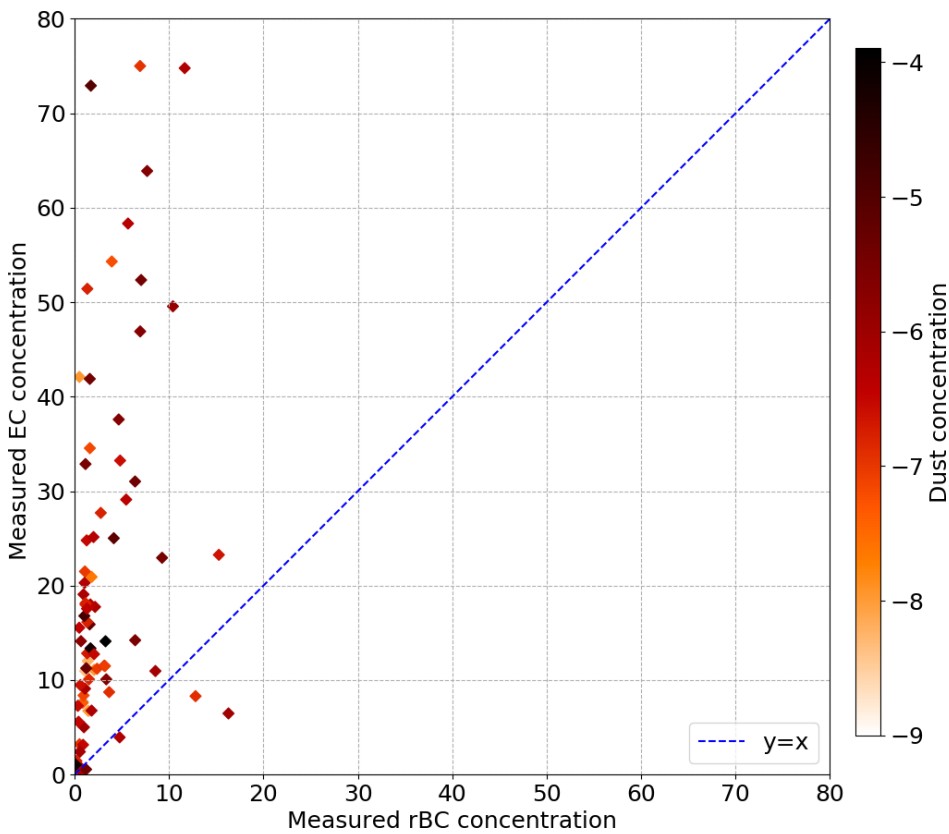

**Figure 4.** Comparison between EC measurements and rBC measurements for all available measurements over the two snow seasons. As the vertical resolution of rBC measurements is higher than for EC measurements, the rBC concentration is computed as the average of all rBC measurements weighted by snow density in the corresponding EC layer.



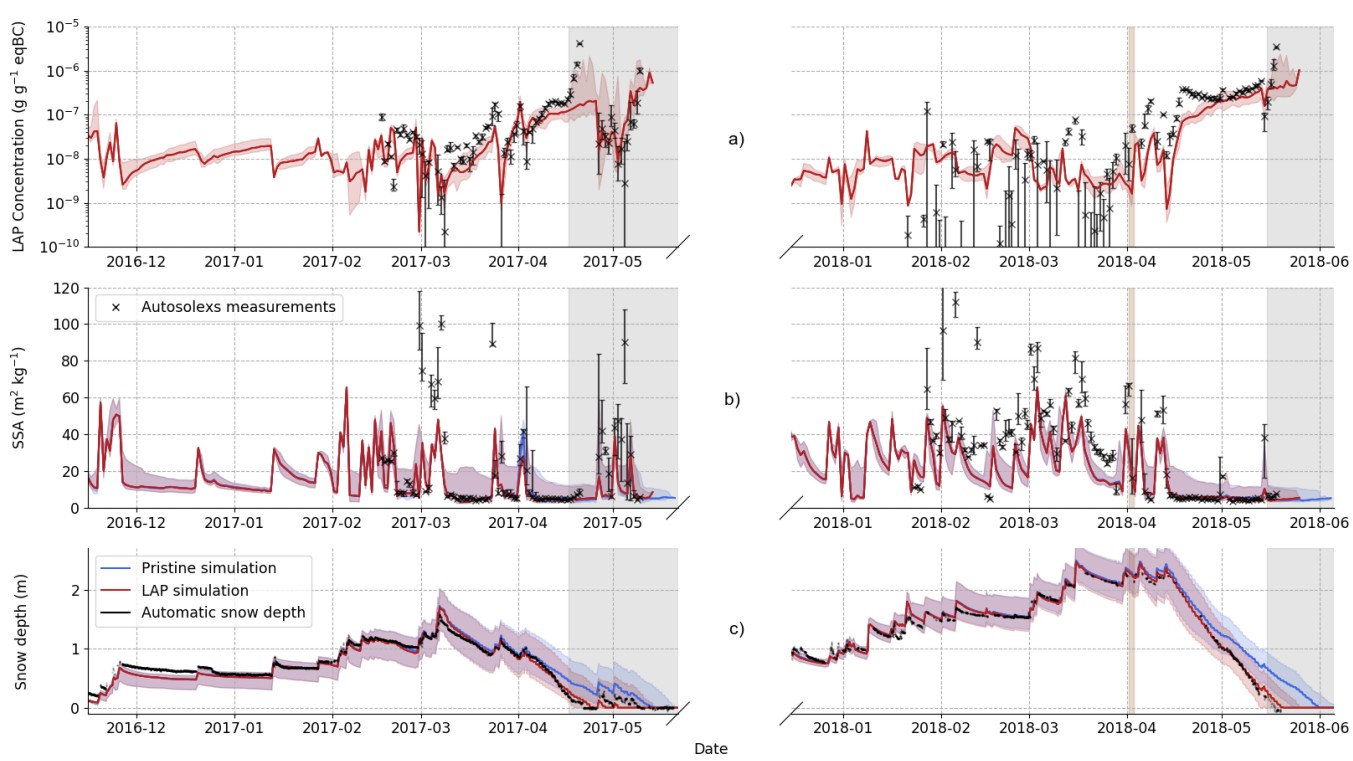

**Figure 5.** Evolution of measured and simulated near-surface LAP concentration (b), near-surface SSA (c) and snow depth (c). Ensemble simulation results are represented by a shadowing and the median value of the different simulations is represented with a full line. The information retrieved from automatic spectral albedo (Autosolexs) are represented by black crosses with error bars corresponding to the 25 and 75% quantile of all the measurements of the day. The major dust deposition of the second year is represented by brown shading and grey shading corresponds to areas with less than 20 cm of measured snow depth.



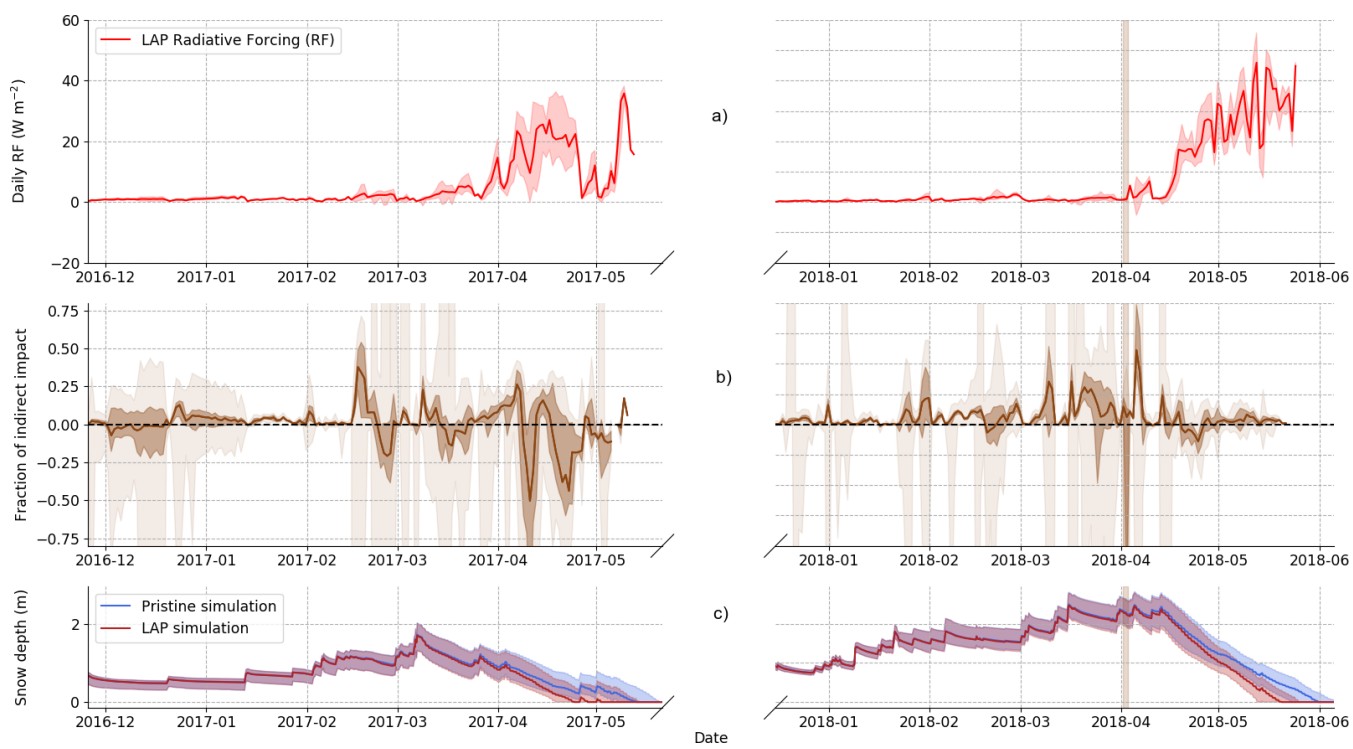

**Figure 6.** a) Evolution of daily radiative forcing (RF) due to LAPs. This RF is computed as the difference of energy absorbed by the snowpack between the LAP and pristine simulations. b) Evolution of the daily fraction of the indirect impact ($r_{ind}$), the opaque shadowing corresponds to the ensemble members between the first and third quantile whilst the light shadowing is the full ensemble spread. c) Simulated snow depth for the pristine and the LAP simulations. The major dust deposition event of the second year is represented by brown shading.



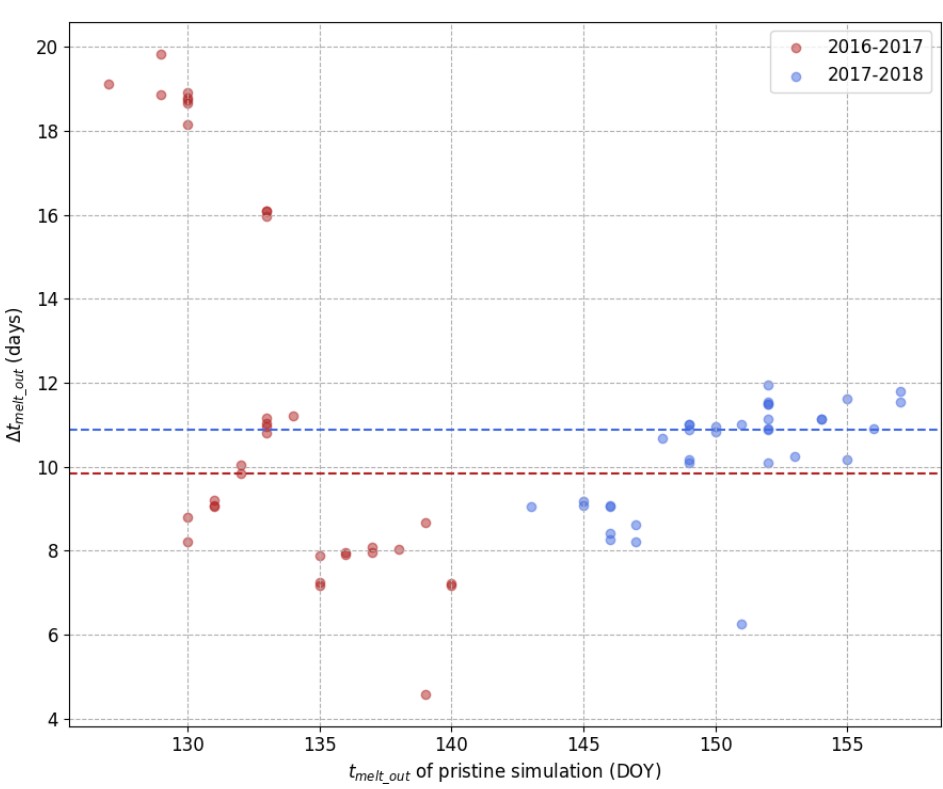

**Figure 7.** $\Delta t_{\text{melt-out}}$ as a function of the date of melt of the pristine simulation for each of the 35 members and each year.
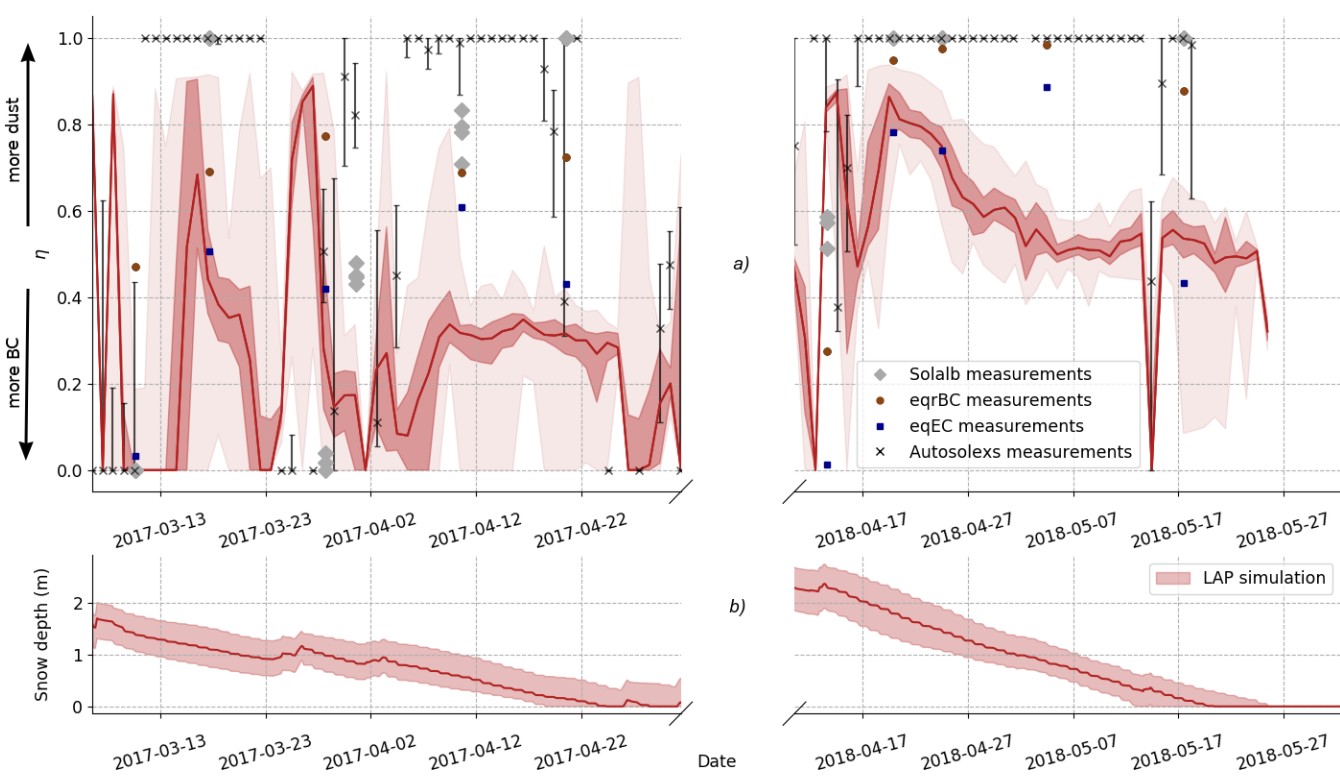

**Figure 8.** a) Evolution of measured and simulated $\eta$ – i.e. the portion of LAP radiative impact caused by dust during the final ablation period. $\eta$ values estimated from the LAP simulation is represented in red, with light shading corresponding to the full ensemble and darker shading corresponding to the first and third quartiles of the ensemble. Information retrieved from automatic spectral albedo are represented by black crosses with error bars corresponding to the 25 and 75 % quartiles of all the measurements of the day. Information retrieved from manual spectral albedo are represented by grey diamonds. Chemical measurements of eqEC (EC+dust) and eqrBC (rBC+dust) are represented in dark blue and brown respectively. The results are put in comparison to the snow depth (b). The major dust deposition event of the second year is represented by brown shading.





## Appendix A: Autosolexs inversion details

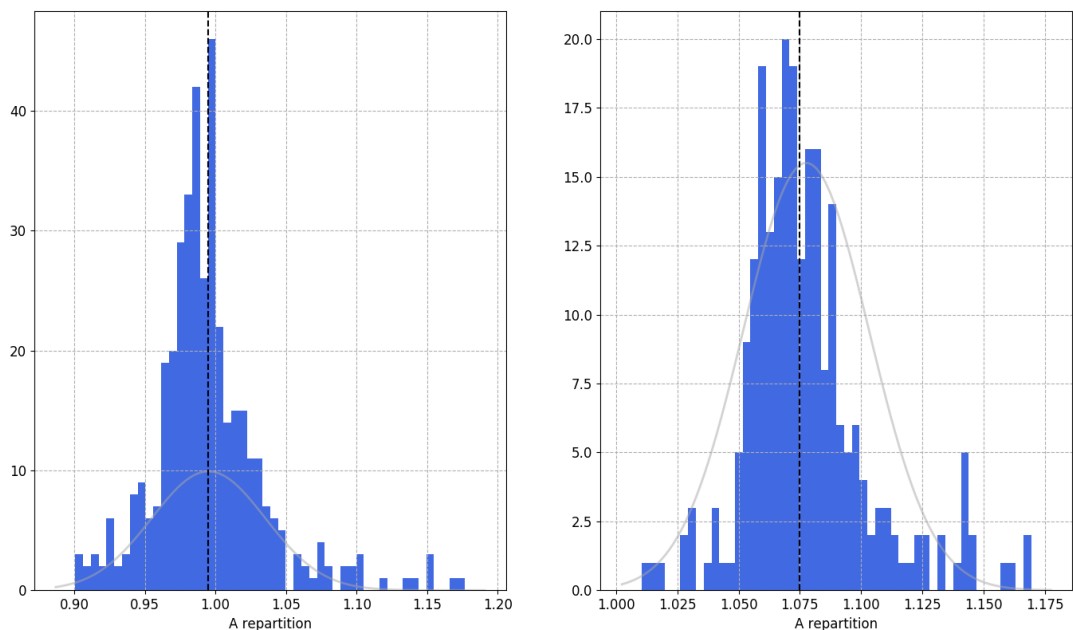

**Figure A1.** Repartition of the scaling factor of Autosolexs measurements for cloudy days. The difference between both years comes from a change in the instrument setup of Autosolexs for the second year; the lower glass protective shield of the light collector was removed.





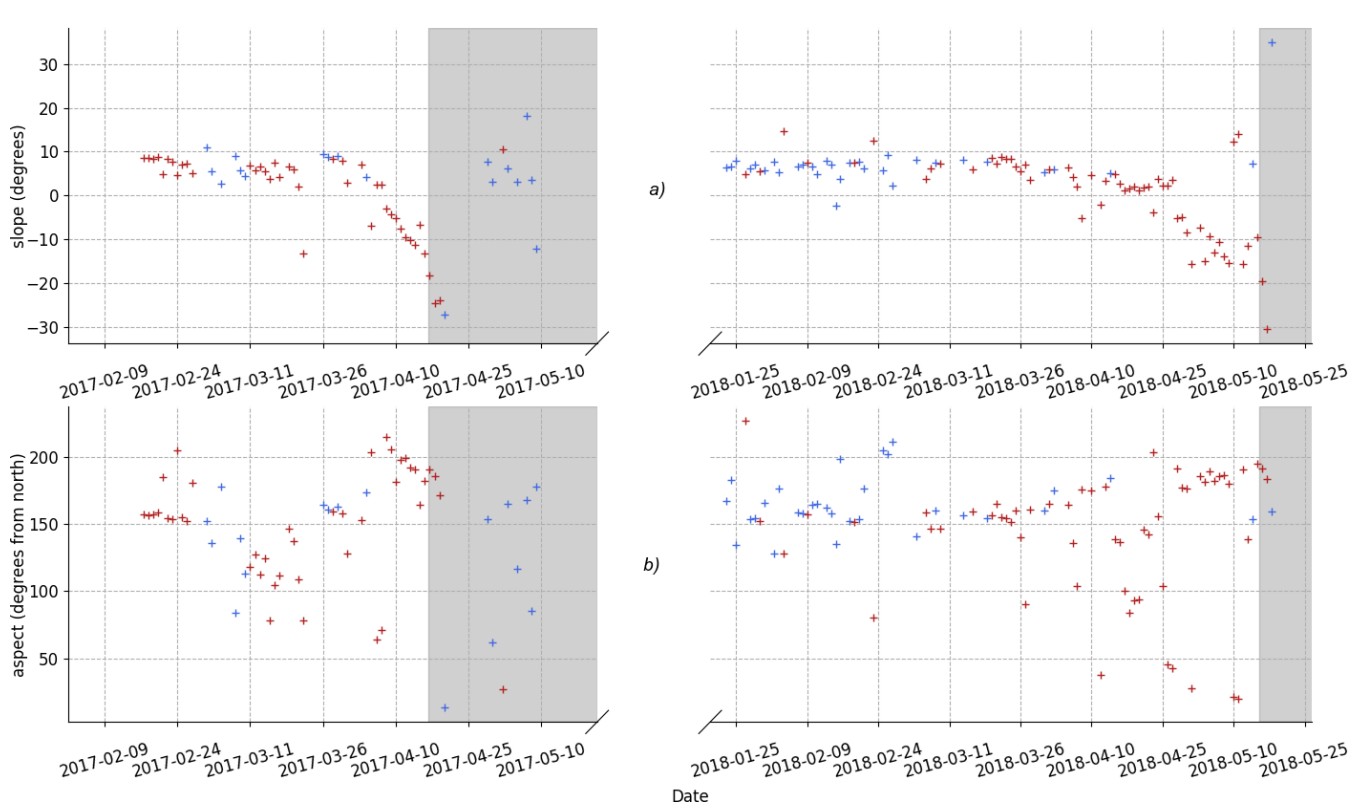

**Figure A2.** Estimation of the slope (a) and aspect (b) of the snow surface under Autosolexs sensor. The estimation is more robust for low solar elevations and low near-surface AEC. Grey shading corresponds to areas with less than 20 cms of measured snow depth.