# Peer review of "Quantification of the radiative impact of light-absorbing particles during two contrasted snow seasons at Col du Lautaret (2058 m a.s.l., French Alps)"

_The Cryosphere, 2019_

## Referee Comment (RC1) · Anonymous Referee #1 · 11 Mar 2020

Summary comment:

This was a worthwhile project which deserves publication, but I found the paper difficult to read.

Major comments:

(1) Figure 4 compares two established methods for measuring black carbon. It shows, for example, that when rBC=5 ng/g, EC can be anywhere from 4 to 75 ng/g. This is disturbing and demands explanation. The very low rBC values from the SP2 make

me skeptical. The authors cite Wendl et al 2014 (on page 8 line 11). However, the Wendl paper discusses the effect of using different nebulizers, so I'm not sure which one was used here. More detail should be given about how the measurements were done, including the expected sampling efficiency of the SP2 set-up.

(2) In places the discussion of a figure disagrees with what the figure actually shows. For example, page 1 line 20 says "LAP concentration and SSA are correctly reproduced". But Figure 5 shows disagreements commonly a factor of 10 for LAP, and a factor of 2 for SSA.

Page 15 line 22-23, describing Figure 5a, says "temporal variations of near-surface AEC are correctly reproduced. But Figure 5a for 2018 shows poor agreement: e.g. for 2018-03 the red line is at ~3 ng/g, but the black crosses (autosolex) show two clusters of values, one an order-of-magnitude higher at ~50 ng/g, and one an order-of-magnitude lower at ~0.5 ng/g.

Page 15 line 22 gives the correlation r2=0.78. Was this correlation done linearly or logarithmically? If it was done linearly, the apparently good correlation will be the result of many points near zero (i.e. 10-9 not distinguished from 10-8) and a few points at the high end (10-6).

In the discussion of Figure 5b, page 15 line 29 says "there is no significant bias between Crocus SSA and the measurements". This figure should be redrawn, plotting SSA on a log scale instead of linear. In the middle of March 2017, the difference is barely distinguishable, but with magnification I can see that the black crosses average ~5.3 m2/kg and the red line ~3.3 m2/kg, indicating a factor-of-1.6 disagreement (and corresponding to effective radii re of 600 and 1000 microns respectively). The same problem would result from changing the vertical axis to be linear in re, because that would shrink the high-SSA region. But there is no reason to prefer either SSA or re as the choice for showing area-to-mass ratio, so to be fair the axis should be in logSSA (or logre); both choices then give the same intervals.

Page 16 line 12 states "the highest AECs estimated from Autosolexs are within the simulated concentrations by our ensemble". I assume AEC means LAP concentration? If so than the statement is not true. If I am reading Figure 5a correctly, the highest AECs from Autosolex (black crosses) are above the red shading, e.g. 2017-03, 2018-03, 2018-04.

Minor comments:

p 7 line 11. "350 to 1050 nm" What is done to account for the rest of the solar spectrum, 300-350 and 1050-2800 nm?

p 11 Eq.3. Instead of requiring the reader to consult Dumont 2017 (which in turn requires going back to Libois 2013 and Picaro 2016), it would help the reader if the authors would describe this equation briefly. For example, why the factor 32/3? And I think B has not been defined.

p 11 line 20. The definitions have been reversed. The density should be rho-ice, not n-ice.

p 12 Eq.4. It would be helpful to describe the psi-function. For example, approximately how much dust would be needed to have the albedo-lowering effect of 1 ng/g BC? Dang et al (JGR 2015) estimated a factor of ∼200 for Saharan dust.

p13 Eq.7. The numerator looks wrong. I think it should instead be E(indirect)-E(pristine).

p 14 line 2. Change "2c" to "2b".

p 15 line 1. Change "50 g-1" to "50 ng g-1".

p 15 line 26. "The dispersion . . . is quite low regarding the median value". I don't understand this phrase; perhaps "regarding" is the wrong word.

p 16 line 17. How can snow cover lower the amount of incoming radiation?

p 25 line 32. This paper has now been published in JGR, so the citation can be up-dated.

Figures 2a and 3c (showing snow depth, with gaps), disagree with Figure 5c (which has no gaps). 2a and 3c will be easier to read if the gaps are filled in.

Figure 2a has two kinds of vertical blue dashed lines. What is the distinction between the bold lines and the faint lines?

Figure 2 caption line 1. Reverse (b) and (c). Wind speed is (b).

Figure 3 caption line 3. "grey diamonds". I don't see any grey diamonds.

Figure 4. Give units for both horizontal and vertical axes.

Figure 5. Do the ticks on the horizontal axis mean the beginning of the month or the middle of the month?

Figure 5 caption line 1. Change "(b)" to "(a)". Change "(c)" to "(b)".

Figure 5 caption is confusing. Line 1 says "measured and simulated near-surface LAP". But no measurements of LAP are actually plotted here. What is plotted are not measured LAP, but rather LAP inferred from albedo.

Figure 6. The horizontal grid lines for the right-hand plots (2018) differ from those in the left-hand plots (2017), indicating a different scale. But the vertical axis has a scale only on the left-hand plot. Add vertical-axis labels to the 2018 plots.

Figure 8 caption last line refers to "brown shading" for the major dust deposition event. I do not see the brown shading.

[Figure]

---

## Referee Comment (RC2) · Anonymous Referee #2 · 17 Apr 2020

The manuscript by Tuzet et al. illustrates an interesting dataset of two years of measurements and modeling at the Col du Lautaret experimental site. The site is quite unique and the analysis of those data represents for sure a step forward in the snow science. The manuscript fits well the aim and scope of TC, but I found it a little hasty in some sections. The BC measurements are unprecedented in the Alps, but the presentation should be modified by comparing the concentrations measured in this manuscript with other publications on this topic. It's also important to present the data with the same units (e.g. ppb or ppm) of other studies, so data can be compared. I suggest to

present dust concentration in ppm and BC concentration in ppb, and directly compare these concentration with other measurements in other mountain chains or ice sheets. I think that some further work is needed before publication in TC.

Some specific comments below.

pg1 ln1. the abstract is way too long. I suggest to shorten it.

pg3 ln32. add "is" between "concentration" and "determined"

pg4 ln18. those mentioned are not "chemical techniques"

pg4 ln21. add more details regarding the radiative impact of dust on snow

pg5 ln10. I suggest to add some discussion also on the paper by Niwano et al. 2012 that made use of SMAP model

ref: Niwano, M., Aoki, T., Kuchiki, K., Hosaka, M., and Kodama, Y. (2012), Snow Metamorphism and Albedo Process (SMAP) model for climate studies: Model validation using meteorological and snow impurity data measured at Sapporo, Japan, J. Geophys. Res., 117, F03008, doi:10.1029/2011JF002239.

Section 2 "Materials". this section includes also several methods. I don't understand why the authors separated material and methods in two sections. I suggest to merge them and to harmonize the content.

pg7 ln25. How did you measure the slope/aspect? What are the uncertainties in these measurements? How these uncertainties impact on the albedo correction?

Section 3. I suggest to add more details on the retrieval methods. The reader is continuosly addressed to other papers from the same group.

pg13 ln1. the RF calculation is here strongly dependent on the simulations. A more useful (and replicable) RF estimation would make use only of Autosolex data. Please add this discussion here or later in the manuscript.

[Figure]

Equation 6. I think it should be E_pristine - E_lap

Section 4. from this section I'm missing a comparison between Autosolex, Solalb and simulated spectral albedo

pg14 ln10. "extreme dust deposition". We still don't know the (climatic) average of dust deposition on snow in the Alps. I suggest to replace "extreme" with "strong".

pg14 ln24. I don't see this regression in the manuscript. it should be added.

Figure 4. Figure 4 is a bit puzzling to me. Units are missing from the axes. From this plot I learn that for rbC <10 (ppb?) all possible values of EC are found in experimental data. Dust color coding is totaly useless since it does not add any information to this presentation. More explaination is needed in the text. Authors may also evaluate to delete this plot and find a better way to present these data.

pg15 ln1. "are lower than 50 gˆ-1 eqBC". I thing that ng is missing from the unit.

Figure 3. SSA variability is not particularly clear. Data are very scattered during the accumulation period. This is due to bad retrieval caused by atmospheric variability? the accumulation period of 2018 shows overall higher SSA values with respect to 2017, why? Please describe here or in the discussion section. Always on Figure 3: revise the label in order to present all data in the plot. In fig3a, the label is missing the autosolex measurements. Fig. 3a also shows an increase of LAP concentration during late April 2017. I suggest to present in the manuscript also the prescribed BC and dust depositions simulated by the model for the two years investigated.

pg15 ln 32. Why 65 mˆ2/kg has been selected as a upper bound for SSA?

Section 4.4. A comparison between TARTES and Autosolex could be interesting here.

pg16 ln22. RF values found in this study should be compared with other studies already published.

pg16 ln33. This is strange. The first year featured higher surface concentration of LAPs

and a stronger shortening of the snow season. Here the authors should try a process-based interpretation of their data. It was BC from the atmosphere? possible input from biomass burning or other emissions? Are there undetected dust events? Giving a look to the albedo spectra may help in the interpetation of LAPs concentration since dust and BC have a different impact on the spectra.

pg17 ln26. please add some references to the last sentence.

pg19 ln15. What is a "numerical outcropping"?

Figure 6. in this figure we only see modeled data. It would be interesting to add also retrieval from autosolex data.

Figure 7. Not particularly informative. I suggest to remove it, and to present average numbers in the text.

Figure 8. not easily understandable. I suggest to think a better way to present these interesting data

Figure A2. Here I don't understand why slope is changing sign during the season. It is very odd and makes me question the slope and aspect retrieval developed by the authors. Are those data somehow validated? It would be also informative to plot the slope-aspect of the underling terrain.

---

## Author Comment (AC3) · 24 Jun 2020

Please find enclosed the tracking of the differences between the original and the revised version of the manuscript as a supplementary pdf file. The "latex diff" tool used has encountered some problems on Section 2 as the section "material" and 'methods" have been merged; we apologize for that. The other modification can be read easily with the deletion in red and the additions in blue

Please also note the supplement to this comment:

[Figure]

https://tc.copernicus.org/preprints/tc-2019-287/tc-2019-287-AC3-supplement.pdf

---

## Author Response (AR1)

**Answer to Anonymous Referee #1**

We would like to thank Anonymous Referee #1 for their extensive analysis of our manuscript which helped us improve our paper. All the comments have been addressed and a point by point response is provided below each comment.

The reviewer's initial comments are reported in black, our answer in blue and the corrections in the paper are highlighted in red. The line numbers which are used in the answers correspond to the new version of the manuscript.

**Summary comment:**

This was a worthwhile project which deserves publication, but I found the paper difficult to read.

**Major comments:**

(1) Figure 4 compares two established methods for measuring black carbon. It shows, for example, that when rBC=5 ng/g, EC can be anywhere from 4 to 75 ng/g. This is disturbing and demands explanation. The very low rBC values from the SP2 make me skeptical. The authors cite Wendl et al 2014 (on page 8 line 11). However, the Wendl paper discusses the effect of using different nebulizers, so I'm not sure which one was used here. More detail should be given about how the measurements were done, including the expected sampling efficiency of the SP2 set-up.

 We agree that only mentioning Wendl et al 2014 can be confusing, so we now explicitly mention the nebulizer used in our case (APEX Q), and reference Lim et al 2014, which describes further evaluation of this particular setup.

Discrepancies between rBC and EC measurements are well documented. We acknowledge that our observed discrepancies are larger than usually observed, and we added a short discussion on that in Section 4.1 p.19 l.23 : "This extra coagulation step is usually not implemented, which may explain why the EC/rBC ratios are higher in our case compared to previous studies. This calls for more systematic comparison of rBC vs EC measurements."

(2) In places the discussion of a figure disagrees with what the figure actually shows. For example, page 1 line 20 says "LAP concentration and SSA are correctly reproduced". But Figure 5 shows disagreements commonly a factor of 10 for LAP, and a factor of 2 for SSA.

 This sentence of the abstract was indeed over-optimistic, it has been replaced as follows : p.1 l.15 : "The temporal variations of near-surface LAP concentration and SSA are most of the time correctly simulated."

Page 15 line 22-23, describing Figure 5a, says "temporal variations of near-surface AEC are correctly reproduced. But Figure 5a for 2018 shows poor agreement: e.g. for 2018-03 the red line is at _3

ng/g, but the black crosses (autosolex) show two clusters of values, one an order-of-magnitude higher at _50 ng/g, and one an order of- magnitude lower at _0.5 ng/g.

Indeed, even if the temporal variations of AEC concentrations are in general captured by the model, there are some periods for which the agreement is poor. This has been underlined.16 l.29 "meaning that temporal variations of near-surface AEC are generally well reproduced despite periods with lower agreement (e.g. March 2018)."

Page 15 line 22 gives the correlation r2=0.78. Was this correlation done linearly or logarithmically? If it was done linearly, the apparently good correlation will be the result of many points near zero (i.e. 10-9 not distinguished from 10-8) and a few points at the high end (10-6).

The correlations are done linearly, which have been indicated in the manuscript in Section 3.2. The logarithmic correlations are indeed lower because of the effect described by the reviewer. The following sentence has been added after the paragraph on AEC measurement description p.16 l.7 :

"It is noteworthy that all the Pearson correlation coefficients for AEC presented in Table \ref{tab:LAP} strongly decrease when the regressions are done logarithmically (not shown). The good linear correlation between the different AEC estimates mainly results from two clusters of points: a lower one, with points around 1 ng g$^{-1}$ that are not distinguished from points around 10 ng g$^{-1}$ and a higher one with points featuring concentrations higher than 50 ng g$^{-1}$. This result is not surprising, as AEC estimations from spectral albedo are expected to have a poor accuracy for low concentrations (< 10 ng g$^{-1}$ approximately, e.g. Warren 2013), explaining the low values of logarithmic correlations."

In the discussion of Figure 5b, page 15 line 29 says "there is no significant bias between Crocus SSA and the measurements". This figure should be redrawn, plotting SSA on a log scale instead of linear. In the middle of March 2017, the difference is barely distinguishable, but with magnification I can see that the black crosses average _5.3 m2/kg and the red line _3.3 m2/kg, indicating a factor-of-1.6 disagreement (and corresponding to effective radii re of 600 and 1000 microns respectively). The same problem would result from changing the vertical axis to be linear in re, because that would shrink the high-SSA region. But there is no reason to prefer either SSA or re as the choice for showing area-to-mass ratio, so to be fair the axis should be in logSSA (or logre); both choices then give the same intervals.

The representation chosen in the original version of the manuscript was indeed clearer for high SSA than for lower ones. Figures 3 and 5 have been redrawn following reviewer's recommendation (they are presented below this comment) and we believe they now provide a clearer overview of the data for both low and high SSA. Moreover, the relative difference between simulated SSA and the black crosses is significant in the middle of March 2017. This is not a systematic bias as the one observed for high SSAs, but it is worth to be underlined in the description p17 l.2 : For SSA lower than 15 m2/kg there is no significant bias between Crocus SSA and the measurements, except for a short period in mid-March 2017

[Figure]

New Figure 3

[Figure]

New  Figure 5

Page 16 line 12 states "the highest AECs estimated from Autosolexs are within the simulated concentrations by our ensemble". I assume AEC means LAP concentration? If so than the statement is not true. If I am reading Figure 5a correctly, the highest AECs from Autosolex (black crosses) are above the red shading, e.g. 2017-03, 2018-03, 2018-04.

By this sentence, the authors meant that the seasonal maxima of Absorption Equivalent Concentration (AEC; at the end of the ablation phase) was close to the simulated AEC ensemble. However this was not clear and simulated AEC are in fact slightly underestimating autosolexs seasonal maximum. This sentence has been reformulated as follows and moved p.17 l.27

During the period with significant RF (April 2017, April and 2018 and May 2018 ), the AECs estimated from Autosolexs measurements are within or slightly above the concentrations simulated by our ensemble (Figure 5). This means that the simulated RF, presented here, are expected to be representative of Autosolexs measurements or slightly underestimated.

**Minor comments:**

p 7 line 11. "350 to 1050 nm" What is done to account for the rest of the solar spectrum, 300-350 and 1050-2800 nm?

The instrument only acquires signals in this range of wavelengths. For SSA and LAP content retrieval, this part of the solar spectrum is sufficient.

p 11 Eq.3. Instead of requiring the reader to consult Dumont 2017 (which in turn requires going back to Libois 2013 and Picaro 2016), it would help the reader if the authors would describe this equation briefly. For example, why the factor 32/3? And I think B has not been defined.

This Section has been modified in order to better explain the core of the spectral albedo model used in the retrieval method. However, the reader may have to read Dumont et al. 2017 in order to understand all technical information related to the retrieval method. Redefining the whole method here would be extremely long and is out of the scope of this study. Finally, the enhancement parameter B has been defined in Section 2.1 defined pg 7 line 5.

p 11 line 20. The definitions have been reversed. The density should be rho-ice, not n-ice.

This mistake has been corrected in the revised version of the manuscript.

p 12 Eq.4. It would be helpful to describe the psi-function. For example, approximately how much dust would be needed to have the albedo-lowering effect of 1 ng/g BC? Dang et al (JGR 2015) estimated a factor of _200 for Saharan dust.

In order to clarify the way the eqBC concentration is computed, the psi function (exactly reproduced from Tuzet et al.2019) has been added in the Appendix A1. As it can be interpreted from this figure, 1 ng/g BC has an albedo-lowering effect 250-290 times stronger than Saharan dust in the range of dust concentrations studied here. Of course, this strongly depends on the hypotheses on dust and BC MAE but gives an order of magnitude

The following sentence has been added in the manuscript p.14 l.9 : Appendix A1 illustrates the hypotheses of BC and dust MAE taken here as well as the psi function. More details about the computation of $C_{eqBC}$ are given in Tuzet et al. 2019 .

p13 Eq.7. The numerator looks wrong. I think it should instead be E(indirect)- E(pristine).

The numerator is indeed E(indirect)- E(pristine), the mistake has been corrected in the manuscript. This does not affect the results, all the results of the manuscript were obtained with the correct formula.

p 14 line 2. Change "2c" to "2b".

Done. p.x l.x, the opposite mistake was also corrected and "2b" became "2c" when mentioning air temperatures.

p 15 line 1. Change "50 g-1" to "50 ng g-1".

Done.

p 15 line 26. "The dispersion . . . is quite low regarding the median value". I don't understand this phrase; perhaps "regarding" is the wrong word.

Indeed, what was meant here was "relative dispersion" and not "dispersion regarding the median value" which was an erroneous formulation. The sentence has been replaced by : "The relative dispersion of near-surface LAP concentrations in the ensemble is moderate, …" p.16 l.32

p 16 line 17. How can snow cover lower the amount of incoming radiation?

In this sentence, snow cover has to be replaced by cloud cover. The mistake has been corrected p.17 l.22: "cloud cover that lowers the amount of incoming radiation"

p 25 line 32. This paper has now been published in JGR, so the citation can be updated.

The reference has been updated.

Figures 2a and 3c (showing snow depth, with gaps), disagree with Figure 5c (which has no gaps). 2a and 3c will be easier to read if the gaps are filled in.

The snow depths plotted on Figure 5c are exactly the same data as Figure 2a and 3c, the gaps seem to have been filled because of the superposition of the model data but it is only a visual artefact. The authors voluntarily decided to let missing data as they are in order not to plot hypothetical information at dates for which we have no measurements.

Figure 2a has two kinds of vertical blue dashed lines. What is the distinction between the bold lines and the faint lines?

As the ROS vertical lines are plotted with transparency, and as the width of the plotted lines are larger than 1 hour, the ROS events appear with bold lines when their duration is higher than 1 hour.

Figure 2 caption line 1. Reverse (b) and (c). Wind speed is (b).

Done.

Figure 3 caption line 3. "grey diamonds". I don't see any grey diamonds.

Indeed, the grey diamonds corresponding to solalb data were not plotted in the original version of the manuscript. This mistake has been corrected as follows :

[Figure]

Figure 4. Give units for both horizontal and vertical axes.

Done

[Figure]

Figure 5. Do the ticks on the horizontal axis mean the beginning of the month or the middle of the month?

For all figures, the horizontal ticks represent the beginning of each month. The following sentence has been added at the end of the legend of Figure 2, which is the first figure to use these temporal ticks : The ticks on the abscissa axis correspond to the first day of each month.

Figure 5 caption line 1. Change "(b)" to "(a)". Change "(c)" to "(b)".

Done

Figure 5 caption is confusing. Line 1 says "measured and simulated near-surface LAP". But no measurements of LAP are actually plotted here. What is plotted is not measured LAP, but rather LAP inferred from albedo.

Based on the terminology defined in the manuscript, LAP concentration has been replaced by AEC in the caption.

Figure 6. The horizontal grid lines for the right-hand plots (2018) differ from those in the left-hand plots (2017), indicating a different scale. But the vertical axis has a scale only on the left-hand plot. Add vertical-axis labels to the 2018 plots.

The yticks were indeed disturbingly positioned in Figure 8 of the original version of the manuscript. After verifying the code, the y-limits were the same on left-hand and right-hand plot so that the data presented already corresponded to the min and max values. The mistake has been corrected in the revised version of the manuscript as follows :

[Figure]

Figure 8 caption last line refers to "brown shading" for the major dust deposition event. I do not see the brown shading.

The extreme dust deposition occurred before the beginning of the selected period. The sentence referring to the brown shading has been removed.

**Answer to Anonymous Referee #2**

We would like to thank Anonymous Referee #2 for his relevant comments pointing out some issues in our manuscript. The comments have been addressed point by point and a detailed response to each comment is provided hereafter.

The reviewer's initial comments are reported in black, our answer in blue and the corrections in the paper are highlighted in red. The line numbers which are used in the answers correspond to the new version of the manuscript.

The manuscript by Tuzet et al. illustrates an interesting dataset of two years of measurements and modeling at the Col du Lautaret experimental site. The site is quite unique and the analysis of those data represents for sure a step forward in the snow science. The manuscript fits well the aim and scope of TC, but I found it a little hasty in some sections. The BC measurements are unprecedented in the Alps, but the presentation should be modified by comparing the concentrations measured in this manuscript with other publications on this topic. It's also important to present the data with the same units (e.g. ppb or ppm) of other studies, so data can be compared. I suggest to present dust concentration in ppm and BC concentration in ppb, and directly compare these concentration with other measurements in other mountain chains or ice sheets.

I think that some further work is needed before publication in TC.

In order to address the first concern of the reviewer on comparisons with other publications, the following table was added to the discussion p.18 l.27 : "Table 2 presents a brief comparison of the surface BC concentration measured in our dataset with results of previous studies in other regions of the world."

| Regions | Typical BC content (ng g$^{-1}$) | References |
|---|---|---|
| Greenland | 0.8-4.5 | Mori et al., 2019; Doherty et al., 2010, Polashenshki et al., 2015 |
| Arctic | 8-60 | Doherty et al., 2010 ; Mori et al., 2019 |
| China | 20-2000 | Wang et al., 2013 ; Ye et al., 2012 |
| North America (including melt) | 5-70 | Doherty et al., 2014 ; Painter et al., 2012 |

| Antarctic Plateau | 0.2-0.6 | Grenfell et al., 1999; Warren et al., 2006 . |
|---|---|---|
| French Alps (including melt) | 0-80 | This study |
| Swiss Alps (including melt) | 0-50 | Gabbi et al., 2015 |

Then, concerning the units used to present dust and black carbon concentration, we agree that many studies use ppm and ppb as units for particle concentrations. However, it is ambiguous for non-experts, as ppb may be understood as nmol mol$^{-1}$ (as is common in atmospheric chemistry), or µg L$^{-1}$ (as is common in hydrology).

The units used throughout the paper use the international unit system, which is the recommended way. Furthermore by choosing the adequate multipliers ng g$^{-1}$ and µg g$^{-1}$, the numbers are the same as with ppb and ppm.

**Some specific comments below**

pg1 ln1. the abstract is way too long. I suggest to shorten it.

The abstract has been shortened to focus on essential information needed by a reader to comprehend the ins and outs of the paper.

pg3 ln32. add "is" between "concentration" and "determined"

Done

pg4 ln18. those mentioned are not "chemical techniques"

Indeed, the word "chemical" has been removed as some of the mentioned techniques are physical (gravimetry, coulter) and others are chemical (mineralogical properties)

pg4 ln21. add more details regarding the radiative impact of dust on snow

This paragraph of the introduction mainly focuses on the measurement techniques of LAP (dust and BC) while more details on the radiative impact of these LAPs in the region of interest can be found in the second paragraph (e.g p.3 l.16)

pg5 ln10. I suggest to add some discussion also on the paper by Niwano et al. 2012 that made use of SMAP model

The radiative model included in SMAP (PBSAM) was already cited but the coupling with the detailed snow model described in Niwano et al. 2012 was not. The logic of the authors was to focus on a model able to compute LAP indirect impacts, and to our knowledge, SMAP has never been used for such an application. However, the innovative coupling between a detailed snow model and a radiative transfer model makes SMAP totally relevant in this section. The citation has been added in the following sentence p5 l.4: "In contrast, estimating the indirect RF of LAPs -- which accounts for the albedo feedbacks, i.e the interaction between LAP impacts and snow metamorphism -- necessitates a coupling between a radiative transfer model and a snowpack model simulating snow metamorphism (e.g. SMAP; Niwano et al. 2012)."

Section 2 "Materials". this section includes also several methods. I don't understand why the authors separated material and methods in two sections. I suggest to merge them and to harmonize the content.

As underlined by the reviewer, some methods to process field measurements and to process numerical simulations were in Section "Materials. As suggested by the reviewer, the two Sections "Materials" and "Methods" have been merged in a "Materials and Method" section which has been re-organised.

pg7 ln25. How did you measure the slope/aspect? What are the uncertainties in these measurements? How these uncertainties impact on the albedo correction?

The slope under the albedometer light collector was measured with a digital inclinometer Level Development **SOLAR-2-15-2-RS232** , which has an accuracy of 0.04 degrees. This inclinometer was fixed to a 3m metal ruler of rectangular section and the ruler was checked to be perfectly straight. A first rough estimation of the direction of maximal slope was done visually . Then, a succession of several measurements were done each 10 degrees around and refined to 5 degrees when approaching the maximal recorded slope. This protocol ensures an aspect measurement with an accuracy better than 5 degrees and a slope angle measurement with an accuracy better than 0.2 degrees. This accuracy is sufficient for an accurate albedo correction (Picard et al., 2020). This paragraph has been modified to explicit the protocol as follows p.9 l.13: Slope inclination and azimuth of the snow surface under the sensor  -- that have to be accounted for in the data processing (Dumont et al. 2017) -- are measured after the acquisition with a digital inclinometer Level Development SOLAR-2-15-2-RS232, which has an accuracy of 0.04 degrees. To do so, the azimuth of the greatest slope was first visually determined. Then a series of measurements every 5 degrees around this direction was performed to find the maximum inclination. This protocol ensures an accuracy better than 5 degrees for the aspect measurement and than 0.2 degrees for slope measurement. This accuracy is sufficient for an accurate albedo correction (Picard et al. 2020).

Section 3. I suggest to add more details on the retrieval methods. The reader is continuosly addressed to other papers from the same group.

This Section has been modified in order to better explain the core of spectral albedo modelling used in the retrieval method. However, redefining the full method presented by Dumont et al. 2017 would be extremely long and is not the purpose of the present study.

Equation 6. I think it should be E_pristine - E_lap

Done

Section 4. from this section I'm missing a comparison between Autosolex, Solalb and simulated spectral albedo

In this study, we choose to compare SSA and AEC for Autosolexs, Solalb and the model instead of the albedo spectra. We rely on the fact that SSA and AEC fully determine (once set the direct-to-diffuse irradiance ratio and the type of impurities) the shape of the spectra for a flat surface, and a comparison wavelength by wavelength would be quite hard to read. We thus believe that it would make the manuscript harder to read to add the comparison of the spectra in addition to comparison of the two parameters retrieved from the spectra, i.e. SSA and AEC.

pg14 ln10. "extreme dust deposition". We still don't know the (climatic) average of dust deposition on snow in the Alps. I suggest to replace "extreme" with "strong".

Done

pg14 ln24. I don't see this regression in the manuscript. it should be added.

The regressions of AEC and SSA have been added in appendix D as follows :

[Figure]

Figure D1 : Comparison between the different estimates of near-surface AEC presented in Section 3.2

[Figure]

Figure D2 : Comparison between the different estimates of near-surface SSA presented in Section 3.2

The following sentence has also been added p.15 l.12

The present Section compares the different estimates of near-surface properties presented in Figures 3 a) and b) and all the correlations can be found in Appendix B.

Figure 4. Figure 4 is a bit puzzling to me. Units are missing from the axes. From this plot I learn that for rbC <10 (ppb?) all possible values of EC are found in experimental data. Dust color coding is totaly useless since it does not add any information to this presentation. More explaination is needed in the text. Authors may also evaluate to delete this plot and find a better way to present these data.

The units have been added on the Figure as follows :

[Figure]

Moreover, the paragraph describing this Figure has been modified in the revised version to bring more explanations about its content: p.15 l.22 … eqEC is almost systematically higher than eqrBC. This bias is explained by the strong discrepancies between both BC measurement techniques illustrated in Figure 4. This figure presents a comparison of EC and rBC concentrations for all available samples, including samples that are not close to the surface. Each point on this scatter-plot corresponds to a snow sample and it appears that EC concentration is almost systematically higher than rBC concentration. Indeed, the ratio EC/rBC has a mean value around 10 and ranges from 0.5 to 30. This means that BC concentrations obtained by thermal-optical method are on average an order of magnitude higher than those obtained by laser-induced incandescence. Moreover, the ratio EC/rBC does not feature a clear relationship with the dust concentration measured in the sample (represented by the color of the points).

pg15 ln1. "are lower than 50 gˆ-1 eqBC". I thing that ng is missing from the unit.

Thank you, the mistake has been corrected

Figure 3. SSA variability is not particularly clear. Data are very scattered during the accumulation period. This is due to bad retrieval caused by atmospheric variability? the accumulation period of 2018 shows overall higher SSA values with respect to 2017, why? Please describe here or in the discussion section. Always on Figure 3: revise the label in order to present all data in the plot. In

fig3a, the label is missing the autosolex measurements. Fig. 3a also shows an increase of LAP concentration during late April 2017.

The stronger variability during the accumulation period is mostly due to :

1. the succession of different precipitation events and
2. the fastest decrease rate of SSA for high SSA than for low SSA.

As suggested by reviewer #1, the SSA is now represented in log-scale which improves the readability during the accumulation period.

SSA values are generally higher during the accumulation period for 2018 season than for 2017 because 2017 was warm and underwent several rain on snow events during the accumulation period as explained p.16 l.14 : "Higher values are generally observed during the second snow season compared to the first one. High SSA values are usually observed for fresh and cold snow (Legagneux et al. 2002), that was rarely present at the surface during the 2016--2017 season owing to the warm and wet meteorological conditions of the season."

The labels of Figure 3 and 5 have been modified to present the data in the first panel in which they are used.

I suggest to present in the manuscript also the prescribed BC and dust depositions simulated by the model for the two years investigated.

The deposition fluxes simulated by ALADIN-Climate have been added in Appendix .B as follows:

[Figure]

Figure B1 : Different component of ALADIN-Climate deposition fluxes used as inputs for Crocus snowpack model. The strong dust deposition event that occured in April 2018 is represented by a brown shading. The different panels correspond to wet and dry deposition fluxes for BC and dust, all expressed in g m$^{-2}$ s$^{-1}$ .

pg15 ln 32. Why 65 mˆ2/kg has been selected as a upper bound for SSA? 0.05mm

For now, the only available metamorphism law based on SSA in Crocus was implemented by Carmagnola et al. 2014 which itself has been adapted from a former metamorphism law based on the optical diameter. In this former metamorphism law, the effective diameter had a low limitation of 0.1mm explaining the upper SSA threshold value of 65 mˆ2/kg. As SSA measurements of fresh snow can be higher than this threshold, it should be adapted in the future, but this is out of the scope of this study.

Section 4.4. A comparison between TARTES and Autosolex could be interesting here.

Figure 6. in this figure we only see modeled data. It would be interesting to add also retrieval from autosolex data.

pg13 ln1. the RF calculation is here strongly dependent on the simulations. A more useful (and replicable) RF estimation would make use only of Autosolex data. Please add this discussion here or later in the manuscript.

As these three comments refer to the same idea, we provide here a common response. The reviewer suggests to also compute RF directly from Autosolexs measurement to remove the dependence on simulations and to present these results in comparison to model data in Section 4.4 and Figure 6. Our choice of only presenting the results using the simulations was guided by the following reasons :

1. Autosolexs does not provide an absolute measurement of incoming and reflected irradiance (in W m-2 nm-1), it is not calibrated for this. The irradiance is expressed in arbitrary units and only an irradiance ratio (as the albedo or diffuse to total ratio) is meaningful and well calibrated. As a consequence, Autosolexs does not provide an estimation of the irradiance, and to convert the measurement into an energy in W m-2, a simulated solar irradiance is required.

2. In addition, in order to be useful for the RF calculation, Autosolexs measurements must be (i) corrected from slope effects and (ii) extrapolated outside the measured spectral range (350-1050 nm) to cover the full solar spectrum. Finally, some modeling would be required to estimate a clean snow albedo needed for the calculation of the additional energy absorbed by the snowpack due to LAPs.

As a consequence of (1) and (2), several extrapolations/simulations are required to convert Autosolexs measured signals into RF. We believe that these steps would add a lot of uncertainty in the estimated RF and that the ones simulated by the snow model are of an equal, if not better accuracy. Indeed, for periods when the RF of LAP is the most significant (April 2017, the end of April 2018 and May 2018 ) the AEC concentrations as well as SSA simulated by our ensemble snowpack simulations are close from the one retrieved from Autosolexs, except at the end of April 2018 where the simulated RF is likely to be slightly underestimated by our model. Note that a RF estimated using only Autosolexs values would also not include indirect radiative forcing.

For the comparison between TARTES and Autosolexs, the reasons are similar. We argue that comparing SSA and AEC is equivalent to comparing spectra, and are much simpler to present and to understand because of the lower dimensionality (i.e. no wavelength)

pg16 ln22. RF values found in this study should be compared with other studies already published.

The recent review of Skiles et al. 2018 makes a full inventory of all RF values found in previous studies. As explained p.20 l.5 "The maximum values of daily and instantaneous RF, around 50 and 200 W m-2 respectively, are in range with maximum values for Europe that have recently been put together in the complete review of Skiles et al. 2018".

pg16 ln33. This is strange. The first year featured higher surface concentration of LAPs and a stronger shortening of the snow season. Here the authors should try a process based interpretation of their data. It was BC from the atmosphere? possible input from biomass burning or other emissions? Are there undetected dust events? Giving a look to the albedo spectra may help in the interpretation of LAPs concentration since dust and BC have a different impact on the spectra.

Indeed, the first year features lower LAP surface concentration and some members of the ensemble simulation exhibit a stronger shortening of the season (though the median shortening is smaller). We believe that this counter-intuitive result is particularly interesting as it underlines that the LAP impact does not only depend on the deposited amount but also on the interaction between snow/ground physics and LAP radiative impacts. This is why this result is further discussed in Section 4.2. The following sentence has been added in the manuscript p.18 l.6: So, the upper estimate of $\Delta t_{melt-out}$ is higher for the first year (20 days) than for the second one (12 days), whereas LAP RF is higher for the second year. This counter intuitive result is further discussed in Section 4.2.1.

As autosolexs estimates of LAP AEC and SSA are consistent with ensemble simulations, we believe that the simulated radiative impact of LAP is of the right order of magnitude. Hence this counter-intuitive result can not be attributed to a LAP deposition issue. Finally, as explained in Section 3.5, the impact of BC and dust are separated in our analysed based on the different impact on the spectra (method p11 l.3)

pg17 ln26. please add some references to the last sentence.

References to Skiles et al. 2019 and Tuzet et al. 2017 have been added after the last sentence.

pg19 ln15. What is a "numerical outcropping"?

We agree that this paragraph was confusing. It has been reformulated to better explain what we call a numerical outcropping, i.e the outcropping of sub-surface layer after the melt of the uppermost layer in the simulation. The new paragraph reads as follows p.20 l.29:

The negative impact, that may be surprising at first, can be explained by the outcropping of a sub-surface layer with a higher SSA than the surface layer when Crocus uppermost layer completely disappears due to melt. In the LAP simulation, LAPs can enhance the melt of the uppermost layer and, in some cases when the underlying layer has a higher SSA, the outcropping of this high SSA layer is occurring earlier than in the pure snow simulations. In this case, the energy absorption can be higher for the pristine simulation (larger surface SSA) than for the indirect simulation (lower surface SSA).

Figure 7. Not particularly informative. I suggest to remove it, and to present average numbers in the text.

According to the authors, this figure is an important result of the manuscript as it underlines the necessity to account for modelling uncertainties when estimating the impact of LAP on snow cover evolution. Presenting median or mean in the text is insufficient to illustrate the difference of dispersion in term of $\Delta_{SAG}$ between the two years within the ensemble simulation, which is a valuable and innovative result discussed in Section 5.1 of the manuscript. However, the authors forgot to cite the Figure 7 in this Section which has been corrected in the revised version p.20 l.6 : Nevertheless, the median advance of melt-out date due to LAP is close for both seasons (Figure 7)

Figure 8. not easily understandable. I suggest to think a better way to present these interesting data

Figure 8 has been modified as follows in the new version of the manuscript. The first sentence of the caption has also been modified to clarify what is represented in the Figure.

[Figure]

Figure 8 : Percentage of the LAP total RF which is caused by dust ($\eta$) during the final ablation period.

Figure A2. Here I don't understand why slope is changing sign during the season. It is very odd and makes me question the slope and aspect retrieval developed by the authors. Are those data somehow validated? It would be also informative to plot the slope-aspect of the underling terrain.

During the winter seasons, the slope and the aspect of the surface under our automatic albedo sensor were not measured in order to avoid any disturbance of the snow surface, hence the exact slope and aspect when the ground is snow covered are unknown. It has been estimated to 7.5 degrees with an aspect of 165 degrees in Picard et al., 2020 (Section 4.2). The slope estimation is expected to have a lower accuracy for high LAP concentrations explaining why the estimate is

affected by large uncertainties, up to lead to negative values. The following sentence has been added p.11 l.9 : : 'It is noteworthy that the slope and aspect estimation presented on Figure B2 have a better accuracy for low surface LAP concentrations, explaining why they are more stable during the accumulation period'. This has a negligible impact on SSA and impurity retrieval, a retrieval with constant slope (7.5, 165 degrees) have been tested with no significant change of the results presented here.

**Modifications of the original manuscript that were not suggested by the referees :**

Following the comment of another member of the community, the authors realized that the term Radiative Forcing (RF) should not be used to describe the radiative impact of impurities in snow. The term Surface Radiative Effect would be preferable, however given the common use of the term RF in the community and the numerous acronyms already defined in the manuscript we decided not to change RF for SRE. This has been clarified just after Table 1  p.6  l.3  "Strictly speaking, what is called 
[revised manuscript text omitted]

RMSE
Bias
$r^2$ | : 23
: 23.3 ng g$^{-1}$ eqBC
: 15.9 ng g$^{-1}$ eqBC
: 0.86 | N
RMSE
Bias
$r^2$ | : 21 (18; no outliers)
: 50.4 (23) ng g$^{-1}$ eqBC
: 27.5 (12.8) ng g$^{-1}$ eqBC
: 0.44 (0.88) | N
RMSE
Bias
$r^2$ | : 12
: 106.7 ng g$^{-1}$ eqBC
: 66.7 ng g$^{-1}$ eqBC
: 0.76 |
| eqEC | | | N
RMSE
Bias
$r^2$ | : 21 (18; no outliers)
: 49.2 (24.7) ng g$^{-1}$ eqBC
: 13.2 (-3.15) ng g$^{-1}$ eqBC
: 0.3 (0.72) | N
RMSE
Bias
$r^2$ | :12
: 96.61 ng g$^{-1}$ eqBC
: 54.9 ng g$^{-1}$ eqBC
: 0.73 |
| Solalb | | | | | N
RMSE
Bias
$r^2$ | : 12
: 71.53 ng g$^{-1}$ eqBC
: 26.48 ng g$^{-1}$ eqBC
: 0.83 |

[revised manuscript text omitted]